# ACE2-like carboxypeptidase B38-CAP protects from SARS-CoV-2-induced lung injury

Tomokazu Yamaguchi[1,16], Midori Hoshizaki[1,2,16], Takafumi Minato[1,16], Satoru Nirasawa[3], Masamitsu N. Asaka[4], Mayumi Niiyama[5], Masaki Imai [6], Akihiko Uda[7], Jasper Fuk-Woo Chan[8], Saori Takahashi [9], Jianbo An[1], Akari Saku[1], Ryota Nukiwa[2,10], Daichi Utsumi [4], Maki Kiso[6], Atsuhiro Yasuhara[6], Vincent Kwok-Man Poon[8], Chris Chung-Sing Chan [8], Yuji Fujino[10], Satoru Motoyama[11], Satoshi Nagata [12], Josef M. Penninger [13,14], Haruhiko Kamada[5], Kwok-Yung Yuen [8], Wataru Kamitani[15], Ken Maeda [7], Yoshihiro Kawaoka [6], Yasuhiro Yasutomi [4], Yumiko Imai [2] & Keiji Kuba [1,16✉]

Angiotensin-converting enzyme 2 (ACE2) is a receptor for cell entry of SARS-CoV-2, and recombinant soluble ACE2 protein inhibits SARS-CoV-2 infection as a decoy. ACE2 is a carboxypeptidase that degrades angiotensin II, thereby improving the pathologies of cardiovascular disease or acute lung injury. Here we show that B38-CAP, an ACE2-like enzyme, is protective against SARS-CoV-2-induced lung injury. Endogenous ACE2 expression is downregulated in the lungs of SARS-CoV-2-infected hamsters, leading to elevation of angiotensin II levels. Recombinant Spike also downregulates ACE2 expression and worsens the symptoms of acid-induced lung injury. B38-CAP does not neutralize cell entry of SARS-CoV-2. However, B38-CAP treatment improves the pathologies of Spike-augmented acid-induced lung injury. In SARS-CoV-2-infected hamsters or human ACE2 transgenic mice, B38-CAP significantly improves lung edema and pathologies of lung injury. These results provide the first in vivo evidence that increasing ACE2-like enzymatic activity is a potential therapeutic strategy to alleviate lung pathologies in COVID-19 patients.

[1] Department of Biochemistry and Metabolic Science, Akita University Graduate School of Medicine, 1-1-1 Hondo, Akita 010-8543, Japan. [2] Laboratory of Regulation of Intractable Infectious Diseases, National Institute of Biomedical Innovation, Health and Nutrition (NIBIOHN), 7-6-8 Saito-Asagi, Ibaraki, Osaka 567-0085, Japan. [3] Biological Resources and Post-Harvest Division, Japan International Research Center for Agricultural Sciences, 1-1 Ohwashi, Tsukuba, Ibaraki 305-8686, Japan. [4] Tsukuba Primate Research Center, NIBIOHN, Hachimandai 1-1, Tsukuba-shi, Ibaraki 305-0843, Japan. [5] Laboratory of Biopharmaceutical Research, NIBIOHN, 7-6-8 Saito-Asagi, Ibaraki, Osaka 567-0085, Japan. [6] Division of Virology, Department of Microbiology and Immunology, Institute of Medical Science, University of Tokyo, 108-8639 Tokyo, Japan. [7] Department of Veterinary Science, National Institute of Infectious Diseases, 1-23-1 Toyama, Shinjyuku-ku, Tokyo 162-8640, Japan. [8] State Key Laboratory of Emerging Infectious Diseases, Carol Yu Centre for Infection, Department of Microbiology, Li Ka Shing Faculty of Medicine, The University of Hong Kong, Pokfulam, Hong Kong Special Administrative Region, China. [9] Akita Research Institute of Food and Brewing, 4-26 Sanuki, Arayamachi, Akita 010-1623, Japan. [10] Department of Anesthesiology and Intensive Care Medicine, Osaka University Graduate School of Medicine, Osaka 565-0871, Japan. [11] Department of Surgery, Akita University Graduate School of Medicine, 1-1-1 Hondo, Akita 010-8543, Japan. [12] Laboratory of Antibody Design, NIBIOHN, 7-6-8 Saito-Asagi, Ibaraki, Osaka 567-0085, Japan. [13] Department of Medical Genetics, Life Sciences Institute, University of British Columbia, 2350 Health Sciences Mall, Vancouver, BC V6T 1Z3, Canada. [14] IMBA, Institute of Molecular Biotechnology of the Austrian Academy of Sciences, 1030 Vienna, Austria. [15] Department of Infectious Diseases and Host Defense, Graduate School of Medicine, Gunma University, Maebashi, Gunma 371-8511, Japan. [16]These authors contributed equally: Tomokazu Yamaguchi, Midori Hoshizaki, Takafumi Minato, Keiji Kuba. ✉email: kuba@med.akita-u.ac.jp

The severe acute respiratory syndrome coronavirus (SARS coronavirus or SARS-CoV) emerged in 2003 as an epidemic of highly contagious and deadly respiratory disease[1–3]. In December 2019, a new coronavirus, SARS-CoV-2 was first identified in a pneumonia outbreak in Wuhan, China, and spread rapidly all over the world in a few months, leading to the current pandemic of Coronavirus Disease 2019 (COVID-19)[4,5]. Infection of this virus causes COVID-19 with respiratory symptoms ranging from mild disease to severe acute lung injury/ acute respiratory distress syndrome (ARDS) and multi-organ failure with high mortality, especially in patients with cardiovascular diseases, diabetes or advanced age[6–8]. Angiotensin-converting enzyme 2 (ACE2) was identified as a receptor for entry of SARS coronavirus into host cells in vitro and in vivo[9,10]. The Spike (S) protein in SARS-CoV-2 shares structural similarities with that of the original SARS coronavirus, and SARS-CoV-2 is also shown to utilize ACE2 as a receptor for cell entry in vitro[4,11,12].

ACE2 was originally described as an enzyme homologous to angiotensin-converting enzyme (ACE) in 2000[13,14]. The renin–angiotensin system (RAS) maintains blood pressure and fluid balance[15,16], while ACE2 has been shown to be an important protective factor in several diseases, including heart failure, myocardial infarction, hypertension, acute lung injury, and diabetes[17]. When the RAS is activated, ACE as a carboxypeptidase cleaves angiotensin I to generate the vasopressor octapeptide angiotensin II (Ang II). ACE2 is a negative regulator of the RAS, which catalyzes degradation of Ang II to Ang 1–7, thereby counterbalancing ACE activity[13,14]. RAS activation has been shown to worsen the pathologies of ARDS/acute lung injury[18]. ACE2 expression is downregulated in the lungs of mice with acute lung injury induced by various causes including acid aspiration, sepsis, and lethal influenza infection[18–20]. Loss of ACE2 augments acute lung injury through Ang II upregulation, whereas treatment with recombinant soluble ACE2, which lacks a membrane-anchored domain, improving lung injury in animal models[18]. ACE2 expression is also markedly downregulated in the lungs of SARS coronavirus-infected mice, and injecting Spike protein of the SARS coronavirus into mice down-modulates ACE2 expression, thereby worsening lung injury via RAS activation[10,18]. Although early studies on SARS-CoV-2 suggested that mRNA expression of ACE2 was upregulated in certain subsets of cells in SARS-CoV-2 infected lungs[21], this upregulation appears to generate a truncated ACE2 isoform that cannot bind to Spike and lacks catalytic activity[22].

Treatment with recombinant soluble human ACE2 protein (rshACE2) was demonstrated to exert beneficial effects in various disease models including heart failure and diabetic nephropathy as well as acute lung injury[18,23,24]. rshACE2 has been developed as a therapeutic for treating patients with ARDS/acute lung injury. In addition, rshACE2 was recently shown to bind to SARS-CoV-2 Spike, thereby suppressing the infection of SARS-CoV-2 in cell lines and human stem cell-engineered organoids in vitro[25]. Thus, rshACE2 functions both as a decoy to inhibit cell entry of SARS-CoV-2 and as an enzyme to protect from lung injury. However, the significance of ACE2 enzymatic activity in COVID-19 remains elusive. We recently identified a bacteria-derived carboxypeptidase B38-CAP as an ACE2-like enzyme[26]. Recombinant B38-CAP protein catalyzes the conversion of Ang II to Ang 1–7 as well as hydrolysis of other ACE2 substrate peptides including des-Arg$^9$-bradykinin, apelin, and dynorphin A, with the same potency as ACE2[26,27]. Treatment with B38-CAP downregulates Ang II levels in mice, thereby antagonizing Ang II-induced hypertension and improving heart failure without overt toxicities[26].

In this study, we aimed to investigate the impact of ACE2 enzymatic activity on acute lung injury induced by SARS-CoV-2 infection. We show that SARS-CoV-2-infection or Spike protein downregulates ACE2 expression. B38-CAP as an ACE2-like enzyme, but not as a decoy for the virus, improves the symptoms of SARS-CoV-2-induced lung injury as well as virus-free Spike-augmented lung injury. These results have important therapeutic implications for the use of rshACE2 or RAS inhibitors to treat COVID-19 patients, for which beneficial outcomes have recently been suggested in clinical studies[28,29].

## Results

**Downregulation of ACE2 protein expression by SARS-CoV-2 infection and Spike protein.** To determine the effects of SARS-CoV-2 infection on ACE2 protein expression in the lungs (Fig. 1a), we infected hamsters with SARS-CoV-2 (UT-NCGM02) via intranasal inoculation[30,31]. At 4 days after infection, while viral Nucleocapsid phosphoprotein (NP) was highly expressed in the lungs, expression of ACE2 protein or mRNA was significantly downregulated in the lungs of SARS-CoV-2-infected hamsters (Fig. 1b, c; Supplementary Fig. 1a), and consistent results were obtained with a different isolate of the virus (HKU-001a) (Supplementary Fig. 1b). We reasoned that reduced ACE2 expression might affect the lung pathologies of SARS-CoV-2 infection as

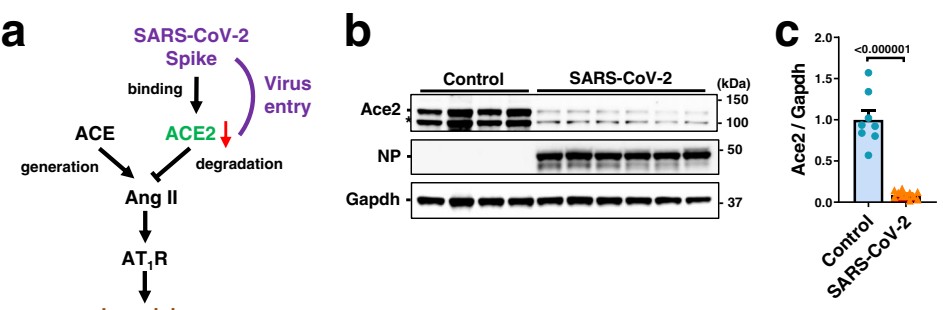

**Fig. 1 Downregulation of ACE2 expression in hamster lung by SARS-CoV-2. a** Schematic drawing of renin-angiotensin system (RAS) and ACE2 in lung injury. Downregulation of ACE2 expression in SARS-CoV-2 infection is implicated by the studies on SARS coronavirus. **b, c** Protein expression of ACE2 in hamster lungs at 4 days after intranasal infection of SARS-CoV-2. Western blot of lung lysates from the hamsters infected with SARS-CoV-2 (UT-NCGM02) (**b**) are shown. NP Nucleocapsid phosphoprotein. * unfolded or short form of ACE2 or non-specific band. Infection with the HKU-001a strain is shown in Supplementary Fig. 1b. Quantification of ACE2 protein abundance (**c**) was done with combined results of the two virus strains; uninfected control ($n = 8$) and SARS-CoV-2 infection ($n = 11$). Values are means ± SEM. Two-tailed unpaired $t$-test. Independent experiments were performed one time (**b**) or two times (**c**), and consistent results were obtained.

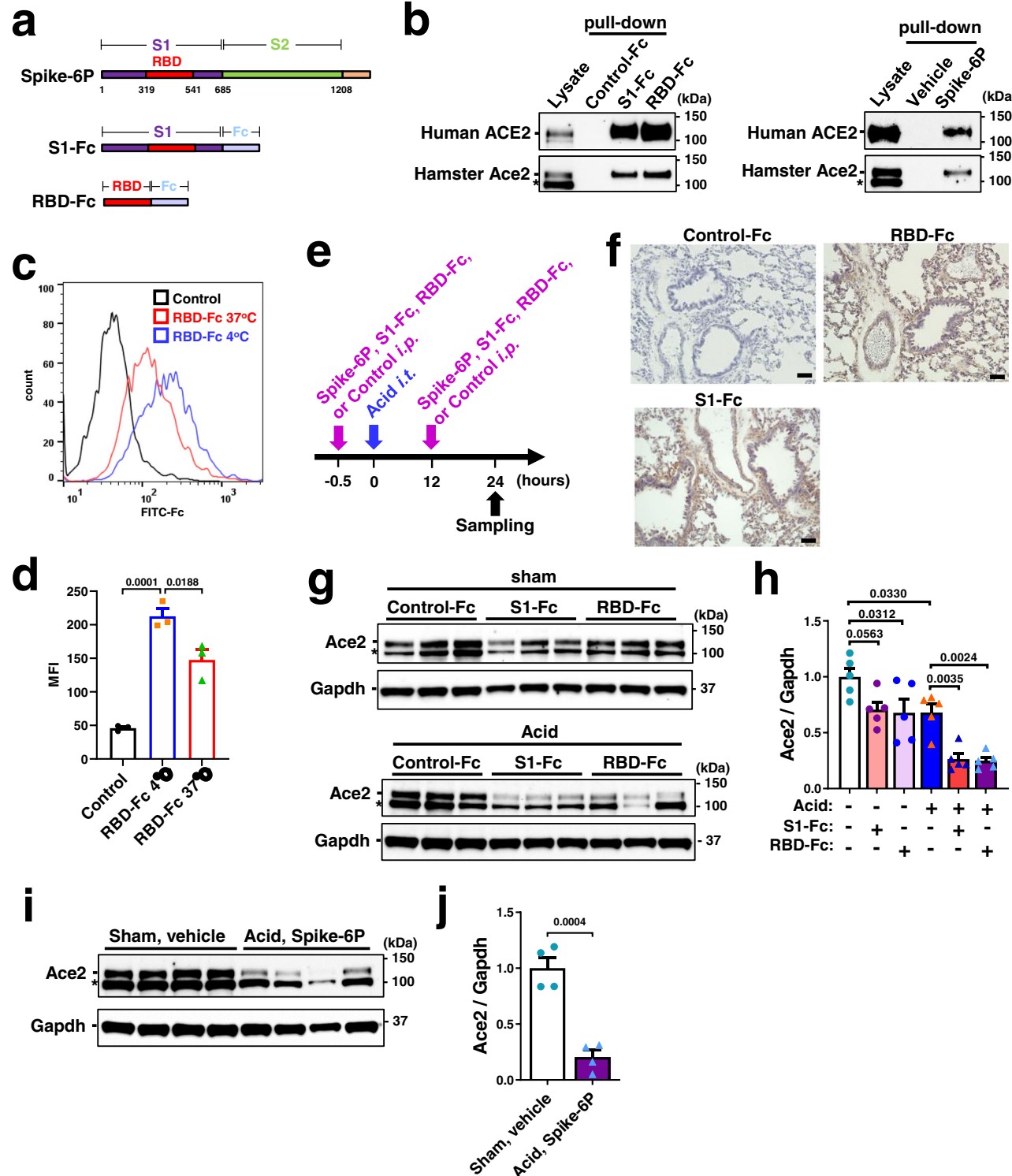

seen in the mice infected with SARS coronavirus[10]. To test this, we prepared three constructs for recombinant proteins of SARS-CoV-2 Spike; trimeric Spike (1–1208 amino acids) with 6 proline substitutions retaining the prefusion conformation[32], S1 domain (1–685 amino acids) and receptor-binding domain (RBD) (319–541 amino acids) fused to human Fc, described hereafter as Spike-6P, S1-Fc, and RBD-Fc, respectively (Fig. 2a). We expressed and purified Spike-6P, S1-Fc, and RBD-Fc proteins in 293 cells (Supplementary Fig. 1c). In in vitro pull-down assays, all the

recombinant Spike proteins indeed bound to both human ACE2 and hamster Ace2 (Fig. 2b; Supplementary Fig. 1d–g). Since recombinant Spike or RBD of the SARS coronavirus down-regulates ACE2 expression[10,33], we examined the effects of SARS-CoV-2 Spike protein on ACE2 expression. We treated Vero E6 cells with RBD-Fc and analyzed expression of endogenous ACE2 on the cell surface. RBD-Fc bound to endogenous ACE2 in Vero E6 cells (Fig. 2c, d; Supplementary Fig. 1h). Decreased binding of RBD-Fc to ACE2 at 37 °C compared with at 4 °C suggested that

**Fig. 2 Recombinant Spike protein of SARS-CoV-2 downregulates ACE2 expression. a** Constructs of recombinant Spike proteins; Spike-6P, S1-Fc, and RBD-Fc. **b** Binding of S1-Fc, RBD-Fc or Spike-6P to human ACE2 and hamster ACE2 in pull-down assays. S1-Fc, RBD-Fc, and Spike-6P but not control-Fc or vehicle pulled down human ACE2 and hamster ACE2 from lysates of Caco2 human colon cells and hamster lungs, respectively. Total lysates are shown as controls. * unfolded or short form of ACE2 or non-specific band. Western Blot of S1-Fc, RBD-Fc or Spike-6P are shown in Supplementary Fig. 1d–g. **c, d** Decreased cell-surface expression of ACE2 after binding to Spike RBD-Fc protein at 37 °C compared to 4 °C in Vero E6 cells. ACE2 surface expression was detected at 3 h of incubation with RBD-Fc using Fc-specific antibody to directly detect surface-bound RBD-Fc and to avoid masking of the ACE2 epitope. Representative FACS histograms (**c**) and quantification of mean fluorescence intensity (**d**) are shown including a background control with an isotype-matched antibody ($n = 3$ per group). FSC/SSC plot was gated for live and healthy Vero E6 cells. One-way ANOVA with Sidak's multiple comparisons test. Numbers above square brackets show $P$ values. **e** Experimental protocol of acid and Spike protein (Spike-6P, S1-Fc or RBD-Fc)-induced lung injury in hamsters. S1-Fc, RBD-Fc, control-Fc (11 nmol/kg for each) or Spike-6P (3.7 nmol/kg) was intraperitoneally injected, and acid was intratracheally instilled (0.1 N HCl, 100 μl per body *i.t.*) under anesthesia. **f** Immunohistochemistry of hamster lungs to detect S1-Fc, RBD-Fc or control-Fc protein using a human Fc–specific antibody. Injected S1-Fc or RBD-Fc but not control-Fc localizes to bronchial epithelial cells, inflammatory cells and alveolar pneumocytes. Bars indicate 100 μm. **g–j** ACE2 protein expression in hamster lungs. Representative Western blot (**g, i**) and quantification of ACE2 protein abundance (**h, j**) are shown. * unfolded or short form of ACE2 or non-specific band. All values are means ± SEM. $n = 5$ hamsters per group and two-way ANOVA with Sidak's multiple comparisons test (**h**) or $n = 4$ hamsters per group and two-tailed unpaired $t$-test (**j**). Numbers above square brackets show significant or nearly significant $P$ values. Independent experiments were performed one time (**b–d**) or two times (**f–j**), and consistent results were obtained.

expression of ACE2 protein on the cell surface was down-regulated by RBD-Fc, partly through internalization or shedding as previously described for SARS coronavirus[33,34] (Fig. 2c, d).

We determined whether SARS-CoV-2 Spike proteins down-regulate Ace2 expression in vivo (Fig. 2e). When hamsters were treated with S1-Fc, RBD-Fc or control-Fc (11 nmol/kg for each), immunohistochemistry with anti-human Fc antibody showed that S1-Fc and RBD-Fc but not control-Fc were localized in the lungs of hamsters (Fig. 2f). In the absence of acid-induced injury, treatment with S1-Fc (11 nmol/kg), RBD-Fc (11 nmol/kg) or Spike-6P (3.7 nmol/kg) did not strikingly affect the abundance of Ace2 protein in the lungs compared with control-Fc or vehicle treatment, though RBD-Fc showed a slight but statistically significant decrease (Fig. 2g, k; Supplementary Fig. 1i, j). On the other hand, when acute lung injury was introduced to hamsters with intra-tracheal instillation of acid (0.1 M HCl, 100 μl) and kept without mechanical ventilation support (Fig. 2e), treatment with S1-Fc, RBD-Fc or Spike-6P significantly down-regulated the abundance of Ace2 protein in the lungs (Fig. 2g–j). Consistently, plasma Ang II levels were significantly upregulated by Spike-6P, S1-Fc or RBD-Fc in the hamsters with acute lung injury but not in the absence of lung injury (Fig. 3c; Supplementary Fig. 2a). Thus, SARS-CoV-2 Spike downregulates ACE2 protein expression in vitro, and Spike treatment plus acid-induced injury downregulates pulmonary ACE2 expression levels and induces RAS activation in vivo.

**SARS-CoV-2 Spike protein worsens acute lung injury.** Without acid instillation the Spike proteins-injected hamsters were apparently healthy. S1-Fc (11 nmol/kg)-treated hamsters induced mild lung edema as defined by the ratio of wet weight to dry weight of the lungs (wet/dry ratio) (Supplementary Fig. 2b, c), whereas treatment with RBD-Fc (11 nmol/kg) or Spike-6P (3.7 nmol/kg) triggered a non-significant increase in the wet/dry ratio (Fig. 3d, e; Supplementary Fig. 2b, c). Histological analysis also showed non-significant mild pathologies in Spike-6P-treated lungs (Fig. 3f, g), though S1-Fc or RBD-Fc treatment exhibited significant increase in lung injury scores (Supplementary Fig. 2d, e). Although the effects of S1-Fc or RBD-Fc may contain acti-vation of immune cells by Fc moiety in the fusion construct (Fig. 2a), Spike protein per se is potentially pro-inflammatory in the lungs.

We next examined the effects of Spike proteins on acid-induced lung injury (Fig. 2e). At 24 h after acid aspiration the lungs appeared mildly reddish and swollen with occasional hemorrhagic spots in control-Fc or vehicle-treated hamsters (Fig. 3d; Supplementary Fig. 2b). When Spike-6P, S1-Fc or RBD-

Fc was intraperitoneally injected in addition to acid instillation, the hamsters exhibited severe lung injury with massive inflam-mation and hemorrhage (Fig. 3d; Supplementary Fig. 2b) and showed a significant increase in the wet to dry weight ratio of the lungs (Fig. 3e; Supplementary Fig. 2c), indicating that Spike protein augmented lung edema. Consistently, histological analysis showed the pathologies of severe lung inflammation in Spike (Spike-6P, S1-Fc or RBD-Fc) plus acid-treated hamsters (Fig. 3f, g; Supplementary Fig. 2d, e). Thus, SARS-CoV-2 Spike protein exacerbates acid-induced lung injury.

**B38-CAP suppresses SARS-CoV-2 Spike-induced lung injury.** Recombinant human soluble ACE2 degrades Ang II while binds to RBD of the Spike thereby inhibiting the entry of SARS-CoV-2 into cells (Fig. 3a). We first examined whether B38-CAP binds to RBD-Fc protein in vitro. While soluble ACE2 was efficiently co-immunoprecipitated with RBD-Fc, B38-CAP was not pulled down with RBD-Fc (Supplementary Fig. 3a, b). In addition, when B38-CAP was added to the in vitro culture of SARS-CoV-2-infected Vero E6/TMPRSS2 cells, viral replication was not affected by B38-CAP (Supplementary Fig. 3c, d). Thus, B38-CAP is an ACE2-like enzyme lacking the ability to inhibit cell entry of SARS-CoV-2. We intraperitoneally injected B38-CAP (2 mg/kg per injection) to the hamsters under acid-induced lung injury with Spike treatment (Spike-6P or RBD-Fc) (Fig. 3b). At 2 h after last injection of B38-CAP or vehicle, ACE2-like enzymatic activity was markedly elevated in the plasma of B38-CAP-treated hamsters (Supplementary Fig. 4a). Measured ACE2 activity may contain the activity of endogenous soluble ACE2 in addition to injected B38-CAP. However, soluble ACE2 is not expressed by alternative splicing in transcription but generated by post-translational mechanisms[35], and B38-CAP does not induce shedding of ACE2 from the cell surface. Thus, we interpreted that the measured ACE2 activity primarily represents the injected B38-CAP level in circulation. In line with increased ACE2 activity, elevated Ang II levels were significantly downregulated in the plasma of B38-CAP-treated hamsters compared with vehicle-treated ones (Fig. 3c; Supplementary Fig. 4b), and Ang 1–7 levels tended to show a slight upregulation (Supplementary Fig. 4c), indicating that injected B38-CAP is functional as an ACE2-like enzyme to degrade Ang II in hamsters. In the hamsters treated with acid only, B38-CAP significantly suppressed acid-induced formation of lung edema and downregulated wet to dry weight ratio of the lungs (Fig. 3d, e; Supplementary Fig. 4d, e). Impor-tantly, in the hamsters under Spike plus acid-induced lung injury, severe inflammation and pulmonary edema was markedly atte-nuated by B38-CAP treatment (Fig. 3d, e; Supplementary Fig. 4d,

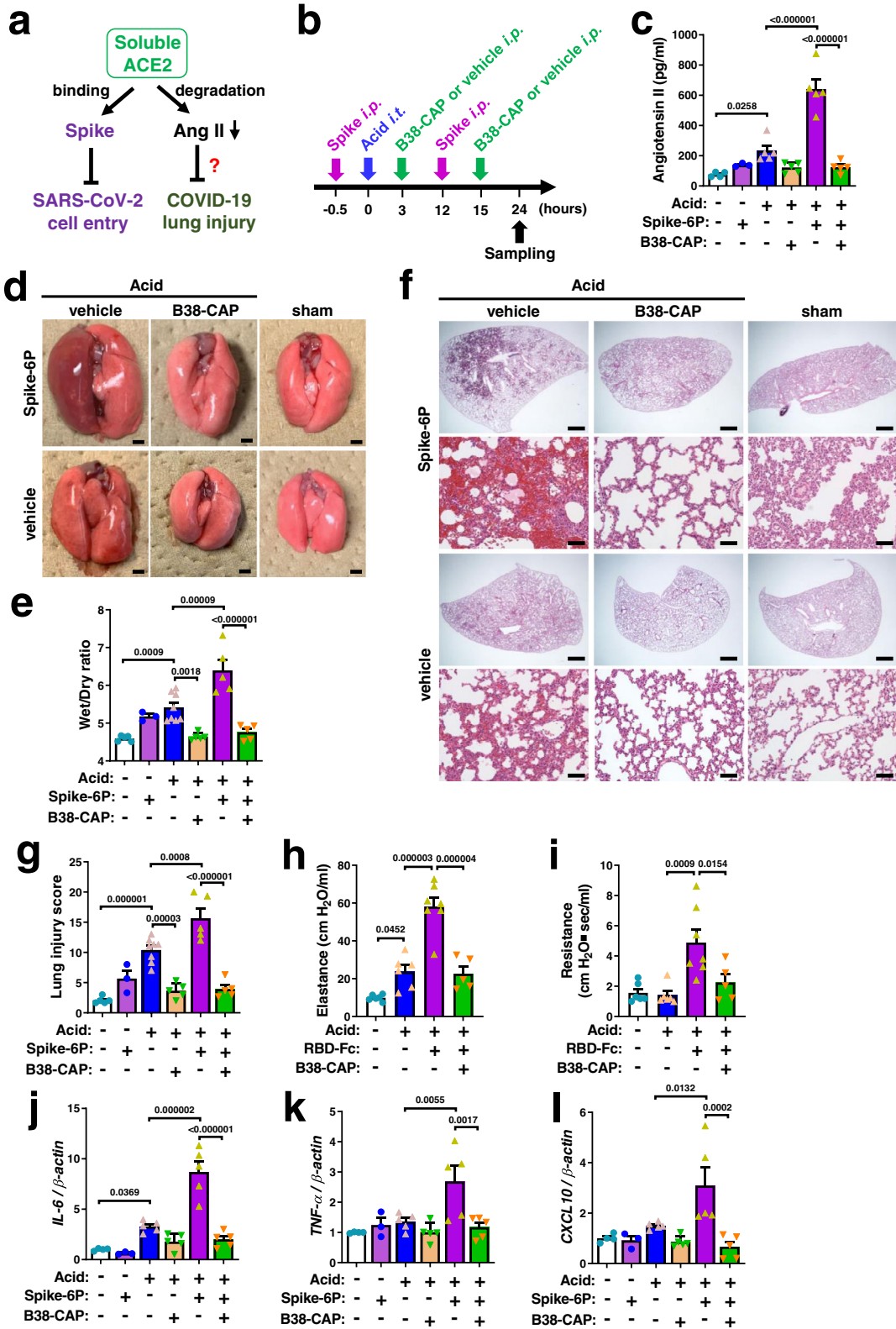

e). Consistently, lung pathologies were significantly improved by B38-CAP (Fig. 3f, g; Supplementary Fig. 4f, g; Supplementary Tables 2 and 3). In addition, lung function was measured as lung elastance and airway resistance. RBD-Fc plus acid-induced impairment of lung function as evidenced by increased

elastance and resistance was significantly improved by B38-CAP treatment (Fig. 3h, i). Furthermore, we examined mRNA expression of cytokines (IL-6, TNF-α, and CXCL10) which are known to be upregulated in COVID-19 patients. While mRNA expression of IL-6, TNF-α, and CXCL10 were also upregulated in

**Fig. 3 Suppression of SARS-CoV-2 Spike-induced lung injury by B38-CAP. a** Effects of soluble ACE2 in SARS-CoV-2 infection and lung injury. **b** Experimental protocol of B38-CAP treatment for hamsters with acid and Spike-induced lung injury. Spike (trimeric Spike-6P protein (3.7 nmol/kg) or RBD-Fc (11 nmol/kg)) or control with or without B38-CAP (2 mg/kg) were intraperitoneally injected (i.p.), and acid (0.1 N HCl, 100 µl per body) was intratracheally instilled (i.t.) under anesthesia. **c** Plasma Ang II measurements at 24 h after acid instillation ($n = 4$ hamsters for sham + vehicle, $n = 3$ for sham + Spike-6P, $n = 5$ each for other experimental groups). **d** Representative photograph of hamster lungs. Bars indicate 2 mm. **e** Wet to dry weight ratios of lungs at 24 h after acid instillation ($n = 3$ hamsters for sham + Spike-6P, $n = 6$ for Acid + vehicle, $n = 5$ each for other experimental groups). **f, g** Lung histopathology. Tissue samples were harvested at 24 h after acid instillation. Representative images are shown (**f**). Bars indicate 1 mm (upper) and 100 µm (bottom) for each treatment. Lung injury score measurements (**g**) (the same experimental cohort as e). **h, i** Lung function measurements. Elastance (**h**) and resistance (**i**) were measured at 17 h after acid instillation ($n = 6$ hamsters each for Sham + control-Fc + vehicle and Acid + control-Fc + vehicle, $n = 7$ for Acid + RBD-Fc + vehicle, $n = 5$ for Acid + RBD-Fc + B38-CAP). **j–l** qRT-PCR analysis of pro-inflammatory cytokine expression in the lungs of hamsters; mRNA levels of IL-6 (**j**), TNF-α (**k**), and CXCL10 (**l**) normalized with β-actin (the same experimental cohort as **c**). All values are means ± SEM. One-way ANOVA with Sidak's multiple comparisons test. Numbers above square brackets show significant P values. Independent experiments were performed two times (**c–l**), and consistent results were obtained.

Spike plus acid-treated hamsters, B38-CAP treatment significantly downregulated high cytokine mRNA levels (Fig. 3j–l; Supplementary Fig. 4h–j). It should be noted that treatment with B38-CAP did not show any toxicities at least in kidney and liver function markers (Supplementary Fig. 4k–n). Therefore, B38-CAP suppresses Spike-induced severe lung injury.

**B38-CAP mitigates lung injury induced by SARS-CoV-2 infection in hamsters.** To address the effects of B38-CAP in acute lung injury induced by SARS-CoV-2 infection, we treated SARS-CoV-2-infected hamsters with B38-CAP. We intratracheally inoculated hamsters with infectious SARS-CoV-2 stock virus at a dosage of $1 \times 10^3$ $TCID_{50}$ per animal. At 4 days post infection (dpi), a high copy number of viral RNA (N gene) was detected in the lungs, and lung edema, pathologies of lung injury, and elevation of cytokine mRNAs were observed (Supplementary Fig. 5a–f). Thus, lung pathologies and cytokine increase were prominent in the hamsters with intratracheal infection. At 12 h after SARS-CoV-2 infection, we initiated the intraperitoneal injection of B38-CAP (2 mg/kg/day) to the hamsters (Fig. 4a). At 4 dpi, body weight decreased to less than 90% in both treatment groups with a further non-significant decrease in B38-CAP-treated animals (Fig. 4b), whereas B38-CAP-treated hamsters were more active and appeared healthier than vehicle-treated ones. The hamsters were euthanized and lung tissues were harvested. When virus N RNA levels were measured as a readout for viral replication in the lungs, copy numbers of N RNA were comparable between B38-CAP treated hamsters and vehicle-treated ones (Fig. 4c), indicating that B38-CAP did not inhibit cell entry of the virus in vivo, similarly to in vitro results (Supplementary Fig. 3c, d). We next quantified Ang II levels in the lung tissues with ELISA. Ang II levels were significantly upregulated upon SARS-CoV-2 infection (Fig. 4d), consistent with downregulation of Ace2 expression (Fig. 1b; Supplementary Fig. 1a), whereas B38-CAP-treated hamsters showed a trend towards a reduction of Ang II levels in the lungs (Fig. 4d), suggesting that B38-CAP is functional as an ACE2-like carboxypeptidase in the infected hamsters. While SARS-CoV-2 infection-induced lung edema was shown by an increased lung weight to body weight ratio (LW/BW) (Fig. 4e) and elevated wet to dry ratio of lung weight (Fig. 4f), B38-CAP significantly improved lung edema as shown by a decreased LW/BW and wet to dry ratio (Fig. 4e, f). In addition, histological analysis consistently showed that B38-CAP improved the lung pathologies (Fig. 4g), and lung injury score was significantly decreased in B38-CAP-treated hamsters compared with vehicle-treated hamsters (Fig. 4h); the scores of thick alveolar wall and hemorrhage were markedly decreased by B38-CAP treatment (Supplementary Table 4). Furthermore, IL-6 mRNA expression was significantly downregulated by B38-CAP treatment (Fig. 4i), and TNF-α mRNA expression also showed a

decreasing trend (Fig. 4j), whereas CXCL-10 expression was not affected (Fig. 4k). Thus, B38-CAP has the ability to suppress inflammation and pathologies in the lungs of hamsters infected with SARS-CoV-2.

**Effects of B38-CAP on SARS-CoV-2-induced lung injury in human ACE2 transgenic mice.** We further investigated the effects of B38-CAP on SARS-CoV-2-induced lung injury by utilizing human ACE2 transgenic (hACE2 Tg) mice, in which expression of human ACE2 is ubiquitously induced by CAG promoter (Supplementary Fig. 6a, b). Human ACE2 protein was detected in the lungs of hACE2 Tg mice but not in wild-type mice (Fig. 5a). We intratracheally inoculated hACE2 Tg mice or wild-type mice with SARS-CoV-2 stock virus (UT-NCGM02) at a dosage of $2 \times 10^3$ $TCID_{50}$ per mouse. All the infected hACE2 Tg mice exhibited a decreased body weight and were dead by 9 days after infection, whereas the virus-inoculated wild-type mice appeared healthy during observation period (Supplementary Fig. 6c, d). Consistently, a high copy number of viral N RNA were detected in the lungs of SARS-CoV-2 infected hACE2 Tg mice (Supplementary Fig. 6e). By contrast, wild-type littermate mice were not apparently infected with the virus, though a low amount of viral RNA was detected in one out of three mice at 2 dpi (Supplementary Fig. 6e), consistent with other recent studies showing no pathologies in wild-type mice in the C57BL/6 background[36,37]. In SARS-CoV-2-infected hACE2 Tg mice, viral RNA (N) was highly detectable in brain, liver, spleen, kidney, and colon as well as in the respiratory system (nasal turbinate, trachea, and lung) (Supplementary Fig. 6f). The kinetics of viral infections varied among the organs. Virus titers and N RNA copy numbers in the lungs peaked at 2 dpi and decreased until day 7 (Fig. 5b; Supplementary Fig. 6e), whereas the virus in the brain became detectable at 4 dpi and markedly increased at day 7, at which point the viral load was the highest among the organs examined (Fig. 5c; Supplementary Fig. 6f), suggesting that SARS-CoV-2 replicates efficiently in brain. As for lung edema, wet to dry ratio of lung weight was significantly increased in SARS-CoV-2 infected lungs of hACE2 Tg mice, peaked at day 2 and day 4 post infection and normalized at day 7, whereas it did not change after virus inoculation in wild-type mice (Fig. 5d). Histopathological analysis showed that lung injury score peaked at 4 days after infection and improved at day 7 in hACE2 Tg mice but not in wild-type mice (Fig. 5e; Supplementary Fig. 7a). Consistently, pulmonary cytokine mRNA levels peaked at day 2 with a sharp decline through day 7, and cytokine levels in the brain markedly elevated at 7 dpi in hACE2 Tg mice but not in wild-type mice (Supplementary Fig. 7b–g). Interestingly, protein expression of both human ACE2 and mouse Ace2 was downregulated in the infected hACE2 Tg mice (Fig. 5f–h). Thus, SARS-CoV-2 infection in these hACE2 Tg mice induces severe lung injury through at 4 dpi, while the mice likely die of explosive viral replication and inflammation in the brain, which is consistent with high lethality of

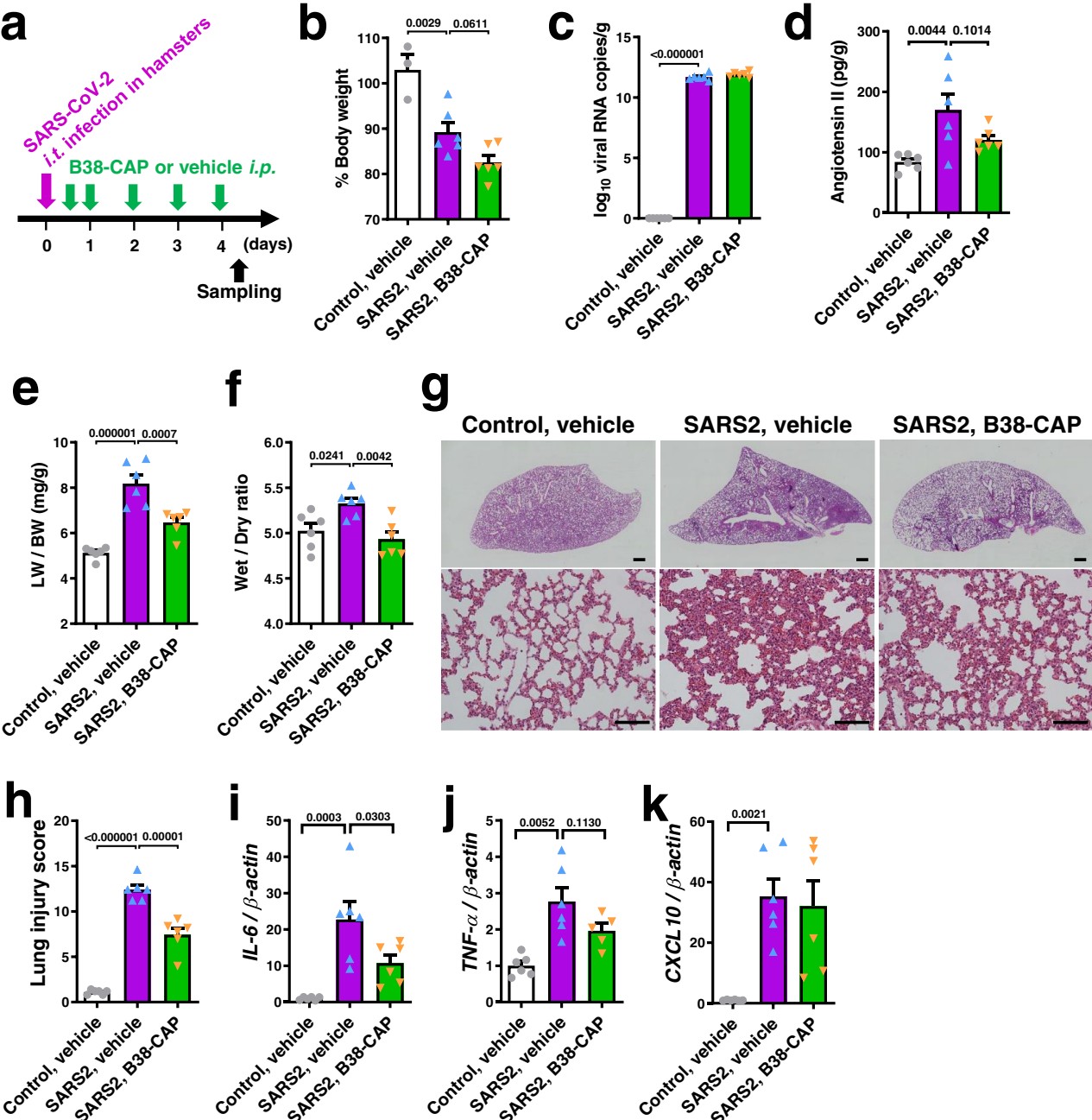

**Fig. 4 B38-CAP suppresses SARS-CoV-2-induced lung injury in hamsters. a** Experimental protocol; Hamsters were intratracheally infected (i.t.) with SARS-CoV-2 ($1 \times 10^3$ TCID$_{50}$), and then the hamsters were injected with B38-CAP (2 mg/kg i.p.) or vehicle once a day. Uninfected control hamsters treated with vehicle ($n = 3$ (b) or $n = 6$ (c–k)), SARS-CoV-2 infected hamsters (SARS2) treated with vehicle ($n = 6$) and SARS2 treated with B38-CAP ($n = 6$) (**b–k**). **b** % Changes of body weight at 4 days after infection from before infection. **c** qRT-PCR of virus N gene expression in the lungs of hamsters. **d** Angiotensin II levels in the lung tissues. **e, f** Lung edema measurements. Lung weight to body weight ratio (**e**) and wet to dry ratio of lung weight (**f**) are shown. **g, h** Lung histopathology. Representative images are shown (**g**). Bars indicate 1 mm (upper) and 100 μm (bottom). Lung injury scores were measured (**h**). **i–k** qRT-PCR analysis of pro-inflammatory cytokine expression in the lungs of hamsters; mRNA levels of *IL-6* (**i**), *TNF-α* (**j**), and *CXCL10* (**k**) normalized with *β-actin*. All values are means ± SEM. One-way ANOVA with Sidak's multiple comparisons test. Numbers above square brackets show *P* values. Independent experiments were performed two times (**b–k**), and consistent results were obtained.

the mice expressing hACE2 in lungs with brain inoculation of SARS-CoV-2 as recently shown[38].

We thus decided to evaluate the effects of B38-CAP on lung injury at 4 days after infection (Fig. 6a). At 4 dpi, the decrease in body weight between B38-CAP-treated mice and vehicle-treated mice was comparable (Fig. 6b), and there was no difference in viral N RNA levels in the lungs between both treatment groups

(Fig. 6c). In accordance with ACE2 downregulation, Ang II levels in the lung tissues were upregulated by SARS-CoV-2 infection (Fig. 6d). By contrast, supplementation of ACE2 activity by B38-CAP injection downregulated Ang II levels (Fig. 6d), whereas Ang 1–7 peptides were not detectable (Supplementary Fig. 8a). B38-CAP significantly attenuated lung edema (Fig. 6e) and the lung pathologies (Fig. 6f), and lung injury score was decreased in B38-

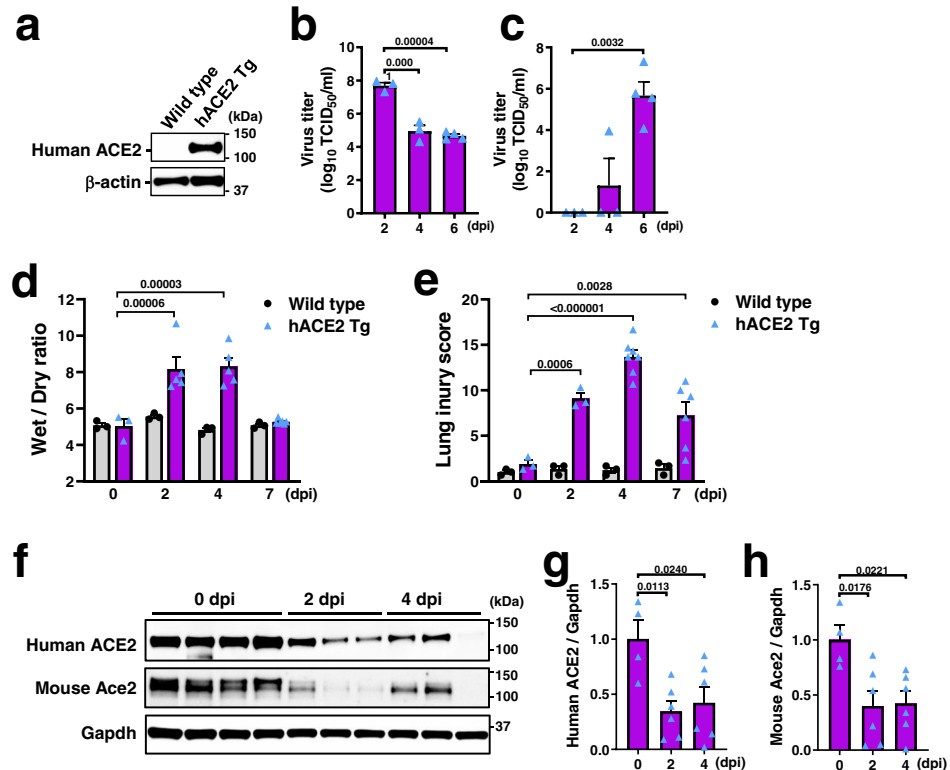

**Fig. 5 SARS-CoV-2-induced lung injury and ACE2 downregulation in hACE2 Tg mice. a** Western blot of human ACE2 protein in the lung lysates of uninfected hACE2 Tg mouse expressing human ACE2 under CAG promoter. **b–c** Virus titers (TCID$_{50}$) in the lysates of lung (**b**) and brain (**c**) prepared from hACE2 Tg mice intratracheally (i.t.) infected with SARS-CoV-2 ($2 \times 10^3$ TCID$_{50}$) at 2 dpi ($n = 3$), 4 dpi ($n = 3$), and 6 dpi ($n = 4$). dpi; days post infection. **d** Lung edema of wild-type and hACE2 Tg mice after SARS-CoV-2 infection. Wet/dry ratio of lung weight of wild type mice ($n = 3$ each for 0, 2, 4, and 7 dpi) and hACE2 Tg mice ($n = 3$ for 0 dpi, $n = 5$ each for 2 and 4 dpi, $n = 6$ for 7 dpi). **e** Lung injury score of wild-type and hACE2 Tg mice after SARS-CoV-2 infection. $n = 7$ for hACE2 Tg mice at 4 dpi, $n = 6$ for hACE2 Tg mice at 6 dpi, and $n = 3$ for other experimental groups. **f–h** Protein expression of human ACE2 and mouse Ace2 in the lungs of SARS-CoV-2-infected hACE2 Tg mice. Representative Western blots are shown (**f**). Human ACE2 (**g**) and mouse Ace2 (**h**) were normalized with Gapdh ($n = 4$ mice for 2 dpi, $n = 6$ each for 2 and 4 dpi). All values are means ± SEM. One-way ANOVA with Sidak's multiple comparisons test (**b–e**, **g**, **h**). Numbers above square brackets show $P$ values. Independent experiments were performed one time (**b**, **c**) or two times (**a**, **d–h**), and consistent results were obtained.

CAP-treated mice compared with vehicle-treated mice (Fig. 6g); the scores of alveolar congestion and wall thickness were significantly decreased in B38-CAP-treated animals (Supplementary Table 5). Increased cytokine expression in the lungs was not significantly improved by B38-CAP treatment, albeit IL-6 mRNA levels showed a decreasing trend in B38-CAP-treated mice (Supplementary Fig. 8b–d). On the other hand, when lung function was measured in the infected hACE2 Tg mice, lung elastance and resistance were significantly decreased in B38-CAP-teated mice compared with vehicle-treated mice (Fig. 6h, i). Thus, B38-CAP improves lung edema and impaired lung functions in hACE2 Tg mice with SARS-CoV-2-induced lung injury.

## Discussion

In this study, we showed that SARS-CoV-2 infection or Spike injection plus acid aspiration decreased the protein abundance of ACE2 in the lungs of hamsters or hACE2 Tg mice, and that Spike downregulated expression of ACE2 protein in the cell culture by itself. B38-CAP was revealed to be an ACE2-like enzyme lacking the structure of a molecular decoy for the virus. We demonstrated that B38-CAP suppressed Spike plus acid-induced lung injury by downregulating Ang II levels, and further found that B38-CAP improved the pathologies of lung injury and impaired lung functions induced by SARS-CoV-2 infection in hamsters and hACE2 Tg mice without affecting viral replication.

SARS-CoV-2 infection decreases both protein and mRNA abundance of ACE2 in the lungs of hamsters. Although early studies of single-cell RNA-sequencing had suggested that interferon signaling upregulates ACE2 mRNA expression thereby potentially augmenting SARS-CoV-2 infection[21], it has been recently shown that interferon and virus induce expression of truncated ACE2 isoform, which is not functional in SARS-CoV-2 binding and carboxypeptidase[22]. Previous studies on SARS coronavirus in 2003 had shown internalization or shedding of cell-surface ACE2 upon viral entry[33,34] as the mechanisms for ACE2 downregulation in the lungs of SARS coronavirus-infected animals[10]. Recently, in the revision of this paper, Spike of the SARS-CoV-2 has been reported to downregulate ACE2 through ubiquitination[39]. On the other hand, downregulation of ACE2 expression had been observed not only in SARS or SARS-CoV-2 infection but also observed in other acute lung injuries caused by acid aspiration, lipopolysaccharide challenge or viral infection[18,20,40]. In our virus-free analyses, cell-surface expression of ACE2 was downregulated by Spike protein in vitro, and both protein and mRNA expression of ACE2 in vivo were downregulated by Spike treatment plus acid-induced injury. Despite a limitation in these analyses due to the absence of virus, the results suggested that there are two modes of ACE2 downregulation in SARS-CoV-2 infection; SARS-CoV-2 binding-mediated ACE2 internalization/shedding[33,34] and lung injury/inflammation-mediated downregulation[18,20,40]. Indeed, Although SARS-CoV-2

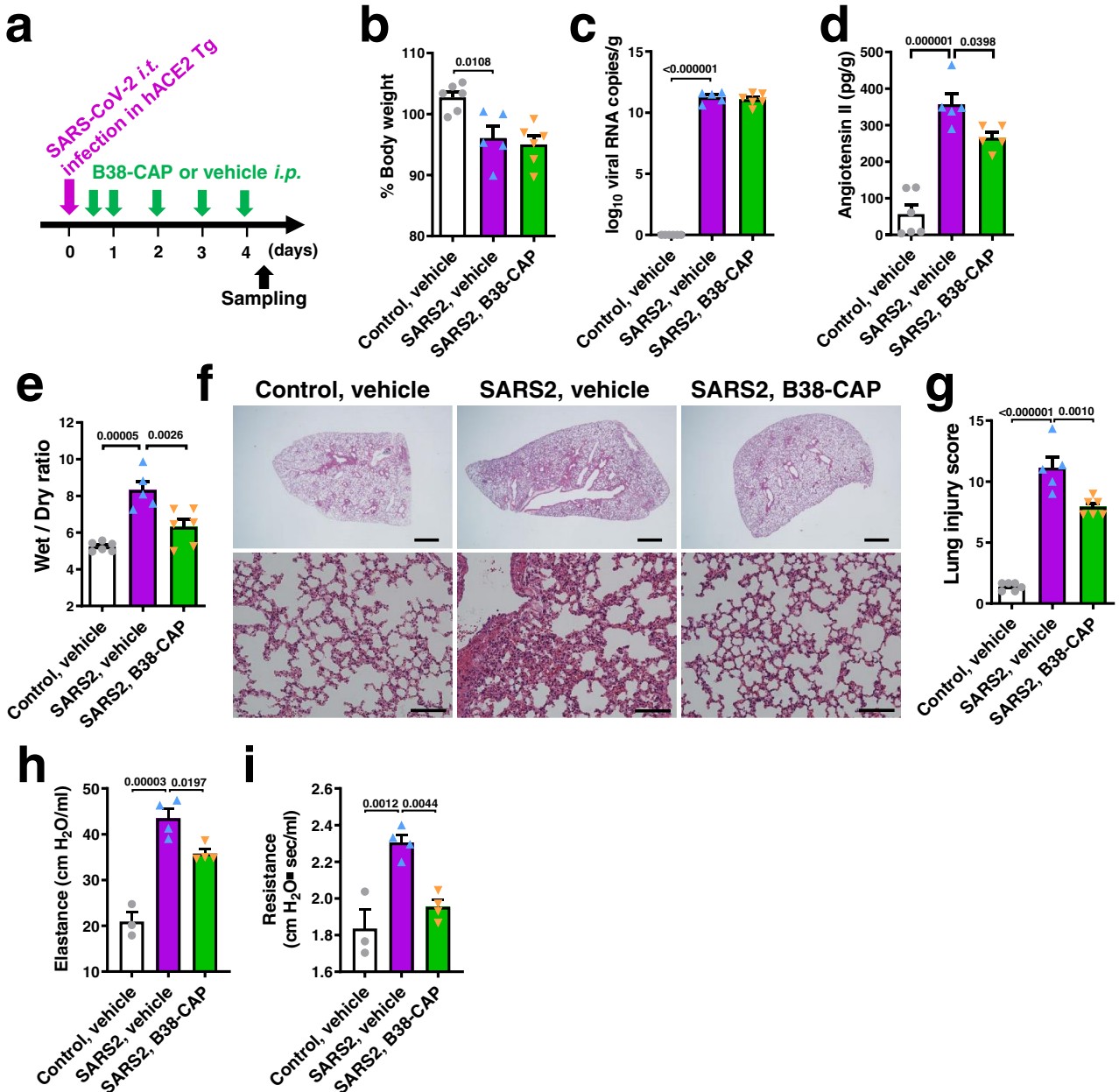

**Fig. 6 B38-CAP improves SARS-CoV-2-induced lung injury and respiratory dysfunction in hACE2 Tg mice. a** Experimental protocol; hACE2 Tg mice were i.t. infected with SARS-CoV-2 ($2 \times 10^3$ TCID$_{50}$), and then the mice were injected with B38-CAP (2 mg/kg i.p.) or vehicle once a day. **b** % Changes of body weight at 4 days after infection from before infection. Uninfected hACE2 Tg mice (control) treated with vehicle ($n = 6$), SARS-CoV-2 infected hACE2 Tg mice (SARS2) treated with vehicle ($n = 5$), and SARS2 treated with B38-CAP ($n = 6$) (**b, c, e, g**). **c** qRT-PCR of virus N gene expression in the lungs of hACE2 Tg mice. **d** Angiotensin II levels in the lung tissues ($n = 6$ mice for control + vehicle, $n = 5$ for SARS2 + vehicle and $n = 5$ for SARS2 + B38-CAP). **e** Wet to dry ratio of lung weight. **f–g** Lung histopathology. Representative images are shown (**f**). Bars indicate 1 mm (upper) and 100 μm (bottom). Lung injury scores were measured (**g**). **h–i** Lung function measurements. Elastance (**h**) and resistance (**i**) were measured at 4 days after SARS-CoV-2 infection in hACE2 Tg mice; control with vehicle treatment ($n = 3$), SARS2 with vehicle ($n = 4$), and SARS2 with B38-CAP ($n = 4$). All values are means ± SEM. One-way ANOVA with Sidak's multiple comparisons test (**b–e, g–i**). Numbers above square brackets show $P$ values. Independent experiments were performed two times (**h–i**) or three times (**b–g**), and consistent results were obtained.

does not bind to mouse Ace2, downregulation of mouse Ace2 as well as human ACE2 was observed in hACE2 Tg mice (Fig. 5e–g). Intriguingly, mRNA expression of mouse Ace2 was also downregulated (Supplementary Fig. 8e, f), suggesting that transcriptional/post-transcriptional mechanism may be involved in the ACE2 downregulation in COVID-19, such as NF-kB-induced microRNA-mediated downregulation of ACE2 mRNA[41]. In relation to the current mRNA vaccine strategies, it should be

noted that the amount of mRNA vaccine-generated Spike protein in the body appears to be lower than that used in our experiments (injection of 1.3 mg/kg of Spike-6P protein in hamsters).

Soluble ACE2 acts via a two-pronged approach to treat COVID-19, as both a carboxypeptidase to suppress lung injury and decoy to neutralize SARS-CoV-2 infection. Since B38-CAP neither binds to RBD of the Spike nor inhibits viral replication in vitro and in vivo, the ACE2-like enzymatic activity of B38-CAP

is crucial for its beneficial effects on lung injury of COVID-19. Interestingly, the beneficial effects of B38-CAP in our COVID-19 lung injury models were observed in the disease measures of lung edema, histopathology, and respiratory function but not to a large extent in cytokine expression. Improvement of lung edema and lung resistance may imply that B38-CAP provides reduction of vascular permeability, water retention, and/or microvascular tones to the microenvironment of SARS-CoV-2-induced lung injury. Accordingly, it is speculated that therapeutics to suppress the RAS, such as angiotensin-receptor blockers (ARBs) and ACE inhibitors, might also be beneficial for COVID-19 patients. In fact, although initial epidemiological studies showed that medication with ARBs or ACE inhibitors did not affect infectivity of SARS-CoV-2 and severity of COVID-19 in the patients[42–44], recent clinical studies have suggested that ARBs or ACE inhibitors provide a good prognosis to the patients[28,29]. The enzymatic activity of both B38-CAP and ACE2 not only degrades Ang II but also generates Ang 1–7, which is protective against lung injury through activation of the Mas receptor[45]. However, detection of low levels of Ang 1–7 suggests that Ang II degradation seems more important in the B38-CAP lung-protective effects than activation of the Mas receptor. On the other hand, both ACE2 and B38-CAP target other peptide as substrates. For instance, des-Arg$^9$-bradykinin is an agonist of B1 bradykinin receptor, and activation of the B1 receptor promotes lung injury[46,47], while ACE2 and B38-CAP degrade des-Arg$^9$-bradykinin[26,27]. Therefore, the beneficial effect of ACE2 enzymatic activity in SARS-CoV-2-induced lung injury might include the effects of degradation of other substrate peptides in addition to Ang II.

Although soluble ACE2 functioning as both a lung-protecting enzyme and decoy for the virus would be ideal for COVID-19 treatment, the efficacy of its decoy function in in vivo infection remains elusive. Focusing on the enhanced conversion of Ang II without Spike binding, such as B38-CAP or soluble ACE2 mutated not to bind Spike, might be an alternative option to treat ARDS in COVID-19 patients. In addition, development of B38-CAP as an inhalation drug might be advantageous because of its cost-effective production and bacterial origin. Most drugs currently available for COVID-19 are repositioned from other diseases and not specifically targeted for COVID-19. Our data show that ACE2 enzymatic activity is a potential therapeutic strategy to alleviate the symptoms of acute lung pathologies in COVID-19 patients, further warranting the significance of specific molecular targeting in treating COVID-19.

## Methods

**Viruses**. SARS-CoV-2 strain (UT-NCGM02) was maintained in Vero 76 cells in Eagle's minimal essential medium (MEM) supplemented with 2% fetal calf serum at 37 °C[31], and virus from passages P3 was used in the experimental infection of hamsters at University of Tokyo. SARS-CoV-2 (Hong Kong strain, HKU-001a (GenBank: MT230904.1)) was isolated from the nasopharyngeal aspirate specimen of a patient with laboratory-confirmed COVID-19 in Hong Kong, plaque-purified and maintained in VeroE6 cells[30]. Virus from passages P3 was used in the infection experiment of hamsters at the University of Hong Kong. For experimental infection and B38-CAP treatment in hACE2 Tg mice, SARS-CoV-2 strain (UT-NCGM02) was obtained from University of Tokyo and propagated in Vero E6/TMPRSS2 (JCRB Cell Bank, JCRB1819) by infecting an MOI of 0.01 and then cultured in DMEM containing 2% FBS at 37 °C for 3 days, and aliquots of the culture media of the virus were stored at −80 °C until use at the Tsukuba Primate Research Center of NIBIOHN. Virus used for animal experiments was from passages P4. For treatment with B38-CAP in SARS-CoV-2 infected hamsters, SARS-CoV-2 (JPN/TY/WK-521) was provided from National institute of infectious diseases, Japan, and propagated and passaged in Vero E6/TMPRSS2 cells[48], and virus of passages P6 was used for experimental infection and B38-CAP treatment in hamsters at Gunma University. All experiments with infectious SARS-CoV-2 were performed in enhanced biosafety level 3 (BSL3) containment laboratories at Tsukuba Primate Research Center of the NIBIOHN, at the University of Tokyo, at the University of Hong Kong or at Gunma University, which followed the approved standard operating procedures of BSL3 facility[30,31].

**Animals**. Three and ten week-old male Syrian hamsters were purchased from SLC Japan and maintained at the animal facilities of Akita University or University of Tokyo. Female Syrian hamsters, aged 6 weeks old, were obtained from the Chinese University of Hong Kong Laboratory Animal Service through the Centre Center for Comparative Medicine Research of the University of Hong Kong. Human ACE2 transgenic (hACE2 Tg) mice, which express human ACE2 in the whole body under CAG promoter, were generated and donated by Drs. Akihiko Uda and Kiyoshi Tanabayashi at National Institute of Infectious Diseases. Human ACE2 cDNA was cloned into pCAGGSP7 vector, and founder mice were generated with pronuclear injection of the expression cassette into zygotes and crossed to C57BL/6 J mice. All three lines of the mice (#5, #16, #17) were infective of SARS coronavirus (Uda A, Tanabayashi K, et al. manuscript in preparation). The mice had been cryo-archived and were recovered at Laboratory Animal Resource Bank of NIBIOHN (https://animal.nibiohn.go.jp/e_ace2_tg_17.html). The recovered line #17 was efficient in expanding colonies and used in this study, and expression of human ACE2 was confirmed with Western blot, and the genotypes of mice were determined by PCR for tail DNA with the primer pair; 5′- CTTGGTGATATGTGGGGTAGA -3′ and 5′- CGCTTCATCTCCCACCACTT -3′. Presence of endogenous mouse Ace2 gene was also confirmed by PCR with the primer pair; 5′- CCGGCTGCTCTTTGA-GAGGACA -3′ and 5′- CTTCATTGGCTCCGTTTCTTAGC -3′. hACE2 Tg mice were maintained as heterozygous (hACE2$^{Tg/+}$) by crossing hACE2$^{Tg/+}$mice with wild-type mice at the animal facility of NIBIOHN or Akita University. Mice were housed in a centrally controlled environment with a 12-h light/12-h dark cycle, temperature of 22–24 °C, and humidity of 30–50%. All animal experiments conformed to the Guide for the Care and Use of Laboratory Animals, Eighth Edition, updated by the US National Research Council Committee in 2011, and approvals of the experiments were granted by the ethics review board of Akita University, NIBIOHN, the University of Tokyo or the University of Hong Kong.

**Experimental infection of hamsters**. To measure ACE2 protein expression in the infected lungs, experimental infection of hamsters was conducted in enhanced BSL3 laboratory at the University of Tokyo and at the University of Hong Kong. Ten-week-old male Syrian hamsters (Japan SLC, Inc.) and 6-week-old female Syrian hamsters (originally from Envigo) were used at the University of Tokyo and at the University of Hong Kong, respectively[30,31]. Under ketamine-xylazine anesthesia, six of 10-week-old male hamsters and five of 6-week-old female hamsters were inoculated via intranasal routes with $10^3$ PFU (in 100 μL PBS) of the SARS-CoV-2 (Tokyo strain, UT-NCGM02) and $10^5$ PFU (in 100 μL PBS) of the virus (Hong Kong strain, HKU-001a), respectively. At four days after infection, hamsters were euthanized with overdose of anesthesia, lung tissues were harvested for preparation of protein lysates and infectious virus inactivated with heat (92 °C, 15 min) and detergent (2% SDS). To investigate the effects of B38-CAP in SARS-CoV-2 infection, 6-week-old male Syrian hamsters (Japan SLC, Inc.) were inoculated via intratracheal routes with $1 \times 10^3$ TCID$_{50}$ (in 100 μL PBS) of the SARS-CoV-2 (JPN/TY/WK-521, passages P6) under anesthesia with cocktails of 0.2 mg/kg medetomidine, 6 mg/kg midazolam and 10 mg/kg butorphanol tartrate in enhanced BSL3 laboratory at Gunma University. Body weight and survival of infected hamsters were monitored for 4 days, and lung tissues were harvested after euthanasia. Cranial lobes of right lungs were used for measurements of wet to dry weight ratio, and the remaining lobes of right lungs were put in RNAlater for RNA extraction and qRT-PCR or were snap frozen and kept at −80 °C until use for peptide extraction for ELISA measurements of Ang II or Ang 1–7. Left lungs were fixed with 4% formalin for histological examination. For B38-CAP treatment, hamsters were infected and divided into two groups. At 12 h post infection, intraperitoneal injection of B38-CAP (2 mg/kg/day) or vehicle was initiated and repeated once a day. At 4 days after infection, the hamsters were euthanized, left and right lungs were harvested as described above.

**Experimental infection of hACE2 Tg mice**. Experimental infection of mice was conducted in enhanced BSL3 laboratory at Tsukuba Primate Research Center of the NIBIOHN. Three to four month-old male hACE2 Tg or littermate wild-type mice were inoculated with $2 \times 10^3$ TCID$_{50}$ of the SARS-CoV-2 (UT-NCGM02, passages P4) in 50 μl PBS via the intratracheal route under anesthesia with cocktails of 0.2 mg/kg medetomidine, 6 mg/kg midazolam, and 10 mg/kg butorphanol tartrate. Body weight and survival of infected mice were monitored and mice showing >25% loss of their initial body weight were defined as reaching experimental endpoint and humanely killed. For assessment of kinetics of infection and pathologies, independent experiments were performed, infected hACE2 Tg or wild-type mice were euthanized, and samples were harvested at 2, 4, 6 or 7 dpi. Thirteen organs (lung, nasal turbinate, trachea, brain, heart, liver, pancreas, spleen, kidney, small intestine, colon, anterior tibial muscle, testis) were harvested from each mouse. Excised organs other than lung and brain were snap frozen and homogenized with TRIzol reagent (Invitrogen) using Beads Grinding Machine (BHA-6; AS ONE) in BSL3 laboratory to determine the viral N RNA copy numbers with qRT-PCR. Cranial lobes of right lungs were subjected to measurements of wet to dry weight ratio, and the remaining lobes of right lungs were put in RNAlater for RNA extraction and qRT-PCR or were snap frozen and kept at −80 °C until use for homogenization for measurements of TCID$_{50}$, protein extraction for Western blot or peptide extraction for ELISA measurements of Ang II or Ang 1–7. Left lungs were fixed with 4% formalin for histological examination. Right cerebral

hemispheres were homogenized for $TCID_{50}$ measurement, and left hemisphere were used for RNA extraction and qRT-PCR. To examine the effects of B38-CAP in SARS-CoV-2 infection, infected mice were divided into two groups, intraperitoneal injection of B38-CAP (2 mg/kg/day) or vehicle was initiated at 12 h post infection and repeated once a day. At four days after infection, mice were euthanized, lung samples were harvested as described above. For measurements of lung function, at 4 dpi the mice were anesthetized, tracheostomized, mechanically ventilated, and lung function measured with Resistance and Compliance system (Buxco). Pulmonary function parameters of elastance and resistance were obtained under mechanical ventilation with tidal volume (10 ml/kg) and PEEP (2 cm $H_2O$) for 15 min.

**Recombinant proteins**. The coding sequence of S1 domain (1–685 amino acids) or RBD (319–541 amino acids) of the Spike protein of SARS-CoV-2 were codon optimized, synthesized, and subcloned into the pCAGGS vector to generate a fusion protein with the Fc portion of human IgG1. Human Expi293F cells (Thermo Fisher Scientific) were transfected with the S1-Fc and RBD-Fc expression vector, supernatants harvested, and S1-Fc and RBD-Fc protein purified by affinity chromatography using a Protein A HP column (Cytiva). Control-Fc protein (#AG714; Merck) was purchased. Trimeric Spike (1–1208 amino acids) with 6 proline substitutions retaining the prefusion conformation (Spike-6P) were prepared as described previously[32]. Recombinant B38-CAP protein was expressed in *E. coli*. The cell lysate was prepared and centrifuged at $13,000 \times g$ for 15 min. The resulting supernatant was subjected to ammonium sulfate precipitation, anion-exchange chromatography with a Q-Sepharose Fast Flow column (1.6 × 10 cm; GE Healthcare), and gel filtration chromatography with a Superdex 75 pg column (2.6 × 60 cm; GE Healthcare). To exclude potential contamination of endotoxin, the eluates were further passed through a Polymyxin B column[26]. All the proteins were confirmed free of endotoxin contamination (<0.1 EU/μg protein) with LAL Endotoxin Assay Kit (Genscript).

**In vitro binding assays**. For pull-down assay to evaluate binding of Spike proteins (S1-Fc and RBD-Fc) to human ACE2 or hamster ACE2 protein, Caco2 human colon epithelial cells or hamster lung tissues were homogenized in lysis buffer (50 mM Tris-HCl, pH 7.4, 1 mM EDTA, 1% NP-40, protease inhibitor (Complete Mini; Roche), 20 mM NaF, and 2 mM $Na_3VO_4$). Protein lysates were incubated with Spike-6P, S1-Fc, RBD-Fc or control human IgG-Fc protein with gentle agitation for 2 h at 4 °C. S1-Fc, RBD-Fc or control-Fc protein were pulled down by Protein G dynabeads (Thermo Fisher Scientific), while Spike-6P protein were pulled down by HIS-Select® Nickel Affinity Gel (SIGMA). Proteins were separated by SDS-PAGE and transferred to a Nitrocellulose membrane. The membranes were probed with anti-human ACE2 antibody (Novus Biological, SN0754, 1:1000 diluted), anti-mouse ACE2 antibody[49] (1:1000 diluted) for detection of hamster ACE2 or anti-human IgG antibody (MBL, 103 R, 1:1000 diluted). For assessment of binding of B38-CAP to RBD-Fc, B38-CAP was detected by anti-B38-CAP polyclonal antibody[26] (1:1000 diluted), and recombinant soluble human ACE2 (R&D systems, 933-ZN) was served as a positive control for binding to RBD-Fc. For flow cytometry analysis to address the effects of RBD-Fc on cell-surface ACE2 expression, Vero E6 cells are detached by 2 mM EDTA/PBS and incubated with RBD-Fc or control-Fc protein at 4 °C or 37 °C for 3 h. Cells were then incubated with FITC-conjugated human IgG-specific polyclonal antibody (Jackson ImmunoResearch, #109-095-088, 1:100 diluted) for detection of RBD-Fc protein and control-Fc bound to Vero E6 cells, and samples were analyzed by flow cytometry using a FACS Calibur (Becton Dickinson)[10,33]. FlowJo software was used to quantify fluorescence intensity.

**Spike protein-mediated acute lung injury in hamsters**. For acid aspiration-induced acute lung injury, three week-old male hamsters were anesthetized with ketamine (200 mg/kg) and xylazine (10 mg/kg), and were intratracheally instilled with 0.1 M HCl (in 100 μl PBS). For spike plus acid aspiration-induced lung injury, SARS-CoV-2 Spike proteins (Spike-6P (3.7 nmol/kg), S1-Fc (11 nmol/kg) or RBD-Fc (11 nmol/kg)) or control-Fc (11 nmol/kg) were intraperitoneally injected 30 min before and at 12 h after acid aspiration. For B38-CAP treatment, B38-CAP protein (2 mg/kg) or vehicle was i.p. injected at 3 and 15 h after acid instillation. To evaluate lung pathologies, the hamsters were euthanized with overdose of anesthesia at 24 h after acid instillation, and blood samples were collected with protease-inhibitor cocktails for Ang II and Ang 1–7 measurements with ELISA and lungs were excised en bloc with hearts. After taking macro photography of the lungs, the left lobe of lungs, which usually exhibit more severe injury than right lungs, was cut into three pieces to measure wet to dry weight ratio, RNA expression and protein expression, and the posterior lobe of right lungs was fixed with 4% formalin samples for histological analysis. For measurements of lung function, the hamsters were anesthetized with ketamine and xylazine at 17 h after acid aspiration, tracheostomized, mechanically ventilated, and lung function measured with Resistance and Compliance system (Buxco). Pulmonary function parameters; elastance, resistance, and dynamic or static compliance were obtained under mechanical ventilation with tidal volume (10 ml/kg) and PEEP (2 cm $H_2O$) for 15 min. After measurements, the hamsters were euthanized with overdose of

anesthesia, and heparinized blood samples were collected for measuring ACE2 enzymatic activity.

**Histopathology and immunohistochemistry**. Lung tissues were fixed with 4% formalin and embedded in paraffin. Five μm thick sections were prepared and stained with Hematoxylin & Eosin (H&E). For semi quantitative assessment of lung injury, the high-resolution images (x200 magnification) of the lung sections stained with H&E were taken with microscope (Nikon). Three randomly chosen fields of each section were scored for Lung injury score in a blinded fashion using a previously defined score consisting of alveolar congestion, hemorrhage, neutrophil infiltration, thickness of alveolar wall, and hyaline membrane formation, as follows: 0 = minimal (little) damage, 1 = mild damage, 2 = moderate damage, 3 = severe damage and 4 = maximal damage[50]. The average lung injury score of three fields of each section were used as the one individual Lung injury score. For immuno-histochemistry of S1-Fc or RBD-Fc proteins in the lungs, 5 μm sections were pretreated with EDTA buffer at 72 °C and stained with peroxidase-conjugated anti-human Fc antibody (Jackson ImmunoResearch, # 109-035-098).

**Western blotting**. Lung proteins were extracted with TNE lysis buffer (50 mM Tris, 150 mM NaCl, 1 mM EDTA, 1% NP40, protease inhibitor (Complete Mini; Roche), 20 mM NaF, 2 mM $Na_3VO_4$) with Microsmash (MS-100R; TOMY), were sonicated and denatured with LDS sample buffer (Invitrogen) at 70 °C. For protein extraction from SARS-CoV-2-infected lungs in BSL3 laboratory, 20 mg pieces of the lung tissues were homogenized by using 3-min Total Protein Extraction Kit (P502L; 101Bio), and the lysates were denatured with 1.3x LDS sample buffer (Invitrogen) or 2% SDS at 92 °C for 15 min. After confirming virus inactivation, the protein samples were taken out from BSL3 laboratory. Proteins were electrophoresed on NuPAGE bis-tris precast gels (Invitrogen) and transferred to nitrocellulose membranes (Invitrogen). Membranes were probed with following antibodies; anti-mouse ACE2 antibody for detection of mouse or hamster ACE2[49] (1:1000 diluted), anti-B38-CAP polyclonal antibody[26] (1:1000 diluted), anti-hamster GAPDH antibody (1:5000; GeneTex, GTX100118), anti-SARS-CoV/SARS-CoV-2 NP antibody[30] (1:20,000 diluted), anti-human ACE2 antibody (R&D systems, MAB9331, 1:250 diluted), anti-β-actin antibody (Sigma, A5316, 1:5000 diluted) or anti-human IgG antibody (MBL, 103 R, 1:1000 diluted). The blotting bands visualized with ECL reagent (Bio-Rad) using ChemiDoc Touch Imaging System (Bio-Rad). Image Lab software was used to quantify band intensity.

**Measurements of cytokine mRNA expression by qRT-PCR**. qRT-PCR analysis was conducted as previously described[26]. Briefly, total RNA was extracted using TRIzol reagent (Invitrogen) and cDNA synthesized using the PrimeScript RT reagent kit (RR037; TAKARA). Quantitative real-time PCR was run in 96 well plates using a SYBR Premix ExTaq II (RR820; TAKARA) according to the instructions of the manufacturer. Relative gene expression levels were quantified by using the Thermal Cycler Dice Real Time System II software (TAKARA). Sequences of the forward and reverse primers of the genes studied are shown in Supplementary Table 1.

**Viral load determination by qRT-PCR**. qRT-PCR to quantify viral N gene copies was conducted in 2 steps; reverse transcriptase (RT) reaction and real-time probe qPCR, with modifications of the diagnostic protocol released from National Institute of Infectious Disease in Tokyo, Japan[51]. The primer and probe sequences are: 5′-AAATTTTGGGGACCAGGAAC-3′ (forward primer (NIID_2019-nCOV_N_F2)), 5′-TGGCAGCTGTGTAGGTCAAC-3′ (reverse primer (NIID_2019-nCOV_N_R2)) and 5′-FAM-ATGTCGCGCATTGGCATGGA-BHQ-3′ (probe (NIID_2019-nCOV_N_P2)). cDNA was synthesized using the Prime-Script RT reagent kit (RR037; TAKARA) in 10 μl RT reaction containing 2 μl 5x PrimeScript Buffer, 0.5 μl PrimeScript RT Enzyme Mix, 10 ng/μl total RNA and 1 μM viral N gene-specific primer (NIID_2019-nCOV_N_R2) with the condition of RT at 42 °C for 15 min and inactivation of reverse transcriptase at 85 °C for 5 s. The RT reaction products were 10-fold diluted, and 1 μl of the diluents were subjected to quantitative real-time PCR using Probe qPCR kit (RR390A; TAKARA) in 25 μl qPCR reaction containing 12.5 μl 2x Premix Ex Taq and 5 μl 5x NIID_2019-nCOV_N_ Primer/Probe mix (XD0007; TAKARA) with 40 cycles of PCR amplification (Denaturing at 95 °C for 5 s; Annealing/Extending at 60 °C for 30 s). The expected amplicon size is 158 bp. The standard curve of qRT-PCR was generated by RT reaction with serial 10-fold dilutions of synthesized N RNA fragment (XA0142; TAKARA) with a known copy number (from $1.0 \times 10^6$ to $1.0 \times 10^2$ copies per μl of RT reaction) in the presence of 10 ng/μl total RNA from uninfected mouse lungs, followed by the real-time probe qPCR with 1 μl of the 10-fold-diluted RT reaction products. These dilutions were used as quantification standards to construct the standard curve by plotting the RNA copy number against the corresponding threshold cycle values (Ct). Results were expressed as log10-transformed numbers of viral N RNA copy numbers per gram of the lung tissues.

**Measurements of Ang II and Ang 1–7 levels and ACE2 activity**. For measurements of plasma Ang II levels, blood samples were collected in tubes containing EDTA (25 mM), o-phenanthroline (0.44 mM), pepstatin A (0.12 mM), and

p-hydroxymercuribenzoic acid (1 mM), and then centrifuged at $1200 \times g$ for 10 min. Angiotensin peptides were acidified and extracted with Sep-Pak cartridges (Waters), and Ang II in the peptide extract was quantified using an ELISA kit (Enzo Life Sciences). Plasma Ang 1–7 concentrations were also determined by subjecting the peptide extracts to measurements with Ang 1–7 ELISA kit (CUSABIO Biotech). To measure the Ang II or Ang 1–7 levels in the lung lysates, frozen lung tissues were homogenized in ice-cold 0.5% formic acid containing 0.5 mg/ml aprotinin and 12.5 mM EDTA, and angiotensin peptides were extracted with Sep-Pak cartridges and the eluates were subjected to ELISA measurements[26]. The measurements for infectious samples were performed in enhanced BSL3 laboratory. For measurements of plasma ACE2 activity, heparinized plasma was diluted with assay buffer, and the reaction mixture contained 40 μl of HEPES buffer, pH 7.5, 0.3 M NaCl, 20 μM Nma-Leu-Pro-Lys(Dnp), 0.01% Triton X-100, 0.02% NaN$_3$, 5 μl diluted plasma, and 5 μl of the HEPES buffer with or without 10 μg/ml MLN-4760, an ACE2 inhibitor in a total volume of 50 μl. The reaction mixture was incubated at 37 °C for 60 min and then the reaction was terminated by adding 0.2 ml of 0.1 M sodium borate buffer pH 10.5, and the fluorescence intensity was measured. Plasma ACE2 activity was determined by subtracting the fluorescence intensity with MLN-4760 from the one without MLN-4760[26].

**Measurement of SARS-CoV-2 replication by TCID$_{50}$.** To measure the viral titer in the lung and brain, the tissues were homogenized in 10 times volumes of the tissue weight of lysis buffer (20 mM Hepes-NaOH (pH7.4), 150 mM NaCl) using Beads Grinding Machine (BHA-6; AS ONE) in BSL3 laboratory. Tissue lysates were serially diluted by a factor of 10 with RPMI1640 containing 5% FBS and penicillin–streptomycin. The diluted lysates were incubated with Vero E6/TMPRSS2 cells ($2 \times 10^4$ cells/well) in 96 well plates for 3 days, and viral titers of each sample were calculated using the Reed–Muench calculation method.

**Neutralization of cell entry and replication of SARS-CoV-2 by using B38-CAP in vitro.** To assess the effects of B38-CAP on cell entry of SARS-CoV-2, both B38-CAP (0, 25, 50 or 100 μg/ml) and SARS-CoV-2 (MOI 0.05) were added to the culture of Vero E6/TMPRSS2 cells ($3 \times 10^4$ cells/well) in 12 well plates for 1 h, and then the cells were washed three times with PBS and incubated with fresh growth medium of RPMI1640 containing 5% FBS and penicillin–streptomycin. For effects of B38-CAP treatment on progeny virus, Vero E6/TMPRSS2 cells were infected with SARS-CoV-2 (MOI 0.05) for 1 h, washed three times with PBS, and incubated with fresh medium containing B38-CAP (0, 25, 50 or 100 μg/ml). Cell lysates were harvested with TRIzol reagent (Invitrogen) at 20 h post infection, and viral N RNA abundance was measured with qRT-PCR.

**Measurements of markers for tissue injury and liver and kidney functions.** Plasma levels of Lactate dehydrogenase (LDH), Aspartate transaminase (AST) or Alanine transaminase (ALT), Creatinine (Cr) and Blood Urea Nitrogen (BUN) were measured as markers of liver and kidney damages, respectively, by using FUJI DRI-CHEM slide kits (Fujifilm corporation).

**Statistical analyses.** Data are presented as mean values ± SEM. Statistical significance between two experimental groups was determined using Student's two-tailed $t$ test. Comparisons of parameters among more than 3 groups were analyzed by one-way ANOVA, followed by Sidak's multiple comparisons test. When a comparison is done for groups with two factor levels, two-way ANOVA with Sidak's multiple comparisons test were used. $P < 0.05$ was considered significant.

**Reporting summary.** Further information on research design is available in the Nature Research Reporting Summary linked to this article.

## Data availability

The data that support the findings of this study are provided in the Article and its Supplementary Information. Source data are provided with this paper. GenBank accession code for SARSCoV-2, HKU-001a is MT230904.1. Source data are provided with this paper.

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

## Acknowledgements

We thank all members of our laboratories for technical assistance and helpful discussions. We thank Dr. Kiyoshi Tanabayashi at National Institute of Infectious Diseases for providing hACE2 Tg mice. K. K. is supported by the Kaken [20H03426, 20K21566] from Japanese Ministry of Science, the FY2020 Program to Develop Countermeasure Technologies against Viral and Other Infectious Diseases [24–136] from Japan Agency for Medical research and Development (AMED) and the Naito Foundation. Y.I. is supported by the Kaken [17H06179], T.Y. is supported by the Kaken [20K07285] from Japanese Ministry of Science and Takeda Science Foundation, J.A. is supported by the Kaken [20K16153] from Japanese Ministry of Science, and J.F.-W.C. is supported by Lo Ying Shek Chi Wai Foundation and the National Program on Key Research Project of China [2020YFA0707500 and 2020YFA0707504]. J.M.P. received funding from the T. von Zastrow foundation, the FWF Wittgenstein award (Z 271-B19), the Austrian Academy of Sciences, the Innovative Medicines Initiative 2 Joint Undertaking (JU) under grant agreement No 101005026, and the Canada 150 Research Chairs Program F18–01336 as well as the Canadian Institutes of Health Research COVID-19 grants F20–02343 and F20–02015.

## Author contributions

T.Y., M.H., T.M., Y.I. and K.K. conceived the study. T.Y., M.H., T.M., J.A., A.S., S.T., Y.F., S.M., J.M.P., Y.I., and K.K. conducted experiments and/or analyzed data. S.Nirasawa, M.N., S.Nagata., and H.K. prepared recombinant proteins. T.Y., M.N.A., M.I., A.U., J.F.-W.C., R.N., D.U., V.K.P., A.Y., C.C.-S.Y., K.-Y.Y., W.K., K.M., Y.K., Y.Y., and K.K conducted infection experiments. K.K. wrote the manuscript with input from all authors.

## Competing interests

J.M.P. is shareholder of Apeiron Biologics which is developing soluble ACE2 (APN01) for COVID-19 therapy. There are no competing interests for other authors.
