## [Peer Review File · Nature Communications]

REVIEWER COMMENTS

Reviewer #1 (Remarks to the Author):

The authors present a manuscript where they aim to show and validate the use of B38-CAP, an enzyme with much similarity to ACE2, in instances of COVID-19.

Major Comments

- It has been shown (<https://doi.org/10.1038/s41586-020-2312-y>) that SARS-CoV-2 causes minimal injury/infiltration in WT mice compared to hACE2 transgenic mice, did you have similar observations between transgenic mice and non-transgenic wildtype controls? Were wildtype controls done for transgenic SARS-CoV-2 infection and rescue with B38-CAP?
- Authors looked at western blot protein expression throughout paper, your non-specific banding is quite pronounced in some westerns and absent in others. Can a more specific antibody be used in place for some westerns?
- Were controls conducted for ACE2 activity assay? If so please show in main figure or supplemental. Since plasma ACE2 activity was calculated based on B38-CAP activity and that there is no way to differentiate ACE2 and B38-CAP activity, how do we know that this bacterial enzyme is just not another variant of ACE2? Authors state in their seminal paper (<https://doi.org/10.1038/s41467-020-14867-z>) that B38-CAP is likely an evolutionary homologue of ACE2, and what has been presented in this paper seems to strongly corroborate that. Authors state that B38-CAP is preventing disease through cleaving Ang II into Ang 1-7, but don't show convincing enough evidence that B38-CAP isn't a simple homologue of ACE2, and similar to soluble rhACE2, binds to the coronavirus preventing disease onset to begin with. The included pull down assay is not convincing enough on its own especially in light of non-specificity in western blotting.
- Why did you test transgenic mice/acid-induced lung injury hamsters with only the RBD-Fc?

Minor comments:

- Your mice/hamsters also had minimal injury with virus in the absence of acid inhalation, is this due to increased ACE2 protein expression with acid treatment? And could increase ACE2 expression due to the acid be leading to more viral infiltration and exuberating of disease severity?
- Be specific on whether you are talking about protein or gene expression throughout the manuscript, please.
- Include control image for lung pathology (Figure 4D).

- No significance shown for decrease in ACE2 (Figure 1I) between control-Fc and S1-Fc (lines 142-143). Please include control-Fc sham group label in Figures.
- In the abstract (lines 45-47) you say that spike protein and the RBD directly elevated Ang II levels when this was not the case (Figure 1J).
- Authors claim to be investigating discrepancy between loss of ACE2 with viral infection versus reports of increased ACE2 mRNA in specific lung cell types, authors need to show mRNA levels and whether they track with their protein expression data to make a conclusive point (lines 86-87).
- It is of note that spike proteins downregulated ACE2, but did not change RAS activation (Ang II), in the absence of acid injury, which ties in with that COVID-19 severity is primarily for those with underlying comorbidities (Figure 1J).

Reviewer #2 (Remarks to the Author):

Review of Minato et al

This paper evaluates the contribution of Angiotensin-converting enzyme 2 (ACE2) activity to the pathogenesis of SARS-CoV-2-induced lung injury using hamster and mouse models. In addition to being the entry receptor for SARS-CoV-2, ACE2 is the carboxypeptidase that degrades angiotensin II to angiotensin 1-7; this attenuates the renin-angiotensin system, which has been shown previously to reduce blood pressure, cardiovascular disease and acute lung injury. Here, the authors perform a series of experiments to address whether the carboxypeptidase enzymatic activity of ACE2 protects against lung injury in hamsters that is induced by SARS-CoV-2 proteins (RBD and S), which downregulate expression of ACE2, or the virus itself in the context of challenge in human ACE2 transgenic mice (expressed under the CAG promoter). Administration of S1-Fc or RBD-Fc directly downregulated ACE2 expression in vivo and resulted in elevated angiotensin II levels and worsened acid-induced lung injury in hamsters. Treatment with B38-CAP, a recombinant protein with ACE2 activity that does not bind S or RBD, downregulated the S1-Fc or RBD-Fc induced increase in angiotensin II, and prevented pulmonary inflammation and disease. The authors went on to test the efficacy of B38-CAP in SARS-CoV-2-infected hACE2 transgenic mice. When B38-CAP was administered starting 12 hours after SARS-CoV-2 infection, lung inflammation and pathology showed improvement at 4 days post-infection (dpi), and this was associated with a small reduction in viral RNA levels even though B38-CAP does not bind S or RBD. The authors conclude that this paper

demonstrates that increasing ACE2 enzymatic activity can be a therapeutic strategy for COVID-19 independent of engagement of the spike protein (which is the basis for some soluble ACE2-Fc therapeutic approaches).

The strengths of this generally easy to follow paper include the quality of the data, the novelty of the approach and inhibitor, the use of two independent animal models, and its potential for translation either directly or indirectly (i.e., using soluble ACE2 with active enzymatic activity or existing drugs that block angiotensin II receptors or inhibit ACE activity). The weaknesses of the study relate to the use of S1-Fc and RBD-Fc and the differences compared to the proteins on virions, the timing of clinical analysis in the hACE2 transgenic mice, and the noticeable absence of virus protection studies in hamsters despite information in the Methods. That said, a few additional controls along with experiments especially in the COVID-19 model could provide needed support for the conclusions of the paper and the potential utility of targeting the renin-angiotensin axis during SARS-CoV-2 infection.

Major Issues.

1. Effects of S1-Fc and RBD-Fc. The authors show in Figure 2 and 3 that administration of S1-Fc and RBD-Fc to hamsters results in down-regulation of ACE2 receptor and increased angiotensin II levels that then correlate with acute lung injury. One major difference between these recombinant proteins and the S on native SARS-CoV-2 virions is the Fc moiety, which could engage immune cells to independently mediate damage and or injury. To confirm these phenotypes are not artifacts of immune cell engagement, the authors should consider generating a LALA-PG or N297Q variant that cannot engage C1q or Fc- gamma receptors and show it mediates the same disease enhancing effects of the parental WT S1-Fc or RBD-Fc. This would also seem important because most advanced vaccine platforms are S protein-based. Is there any evidence from early clinical trials of the vaccines exacerbating lung injury in those with pre-existing pulmonary conditions?

2. Hamster challenge with SARS-CoV-2. The authors perform all preliminary studies with S1-Fc and RBD-Fc in hamsters and yet do not show treatment and challenge data with B38-CAP in these animals. Based on the description in the Methods section (p.15, middle) it appears that hamsters were indeed infected with 10^3 PFU of SARS-CoV-2 UT-NCGM02) and 10^5 PFU of SARS-CoV-2 HKU-001a. Where is this data? This should be included as it directly links to the studies in Figures 2 and 3.

3. CAG-hACE2 transgenic mice. In Figure 4, the authors treated SARS-CoV-2 infected transgenic mice with B38-CAP beginning at 12 hours post-infection and then harvested tissues at day+4. There are several issues: (a) The CAG-hACE2 transgenic mice have not been described for SARS-CoV-2 infection and pathogenesis. While the authors allude to a paper in preparation (p.14, bottom), for the reader

to understand this model, some data needs to be added to this paper (e.g., weight loss and survival; kinetics of viral infection in lungs). (b) The choice of pathological and immunological evaluation for a drug at day+4 is unusual given that many other hACE2-transgenic mice show worse disease at later time points (e.g., day+7 or +8 [PMID:32841215, 32839612, 32516571]). What happens later? Does B38-CAP prevent lethality or severe clinical disease in this CAG-hACE2 transgenic mice? This is an important test of a drug. (c) A more thorough investigation of disease measures, as exemplified by Figure 3, is warranted in the COVID animal model. Some additional measures would greatly enhance the conclusions.

4. In Figure 4c, the authors show that B38-CAP appears to lower viral burden. How does this work since it neither binds to S or RBD? Does B38-CAP down-regulate ACE2 expression through some type of negative feedback? The authors start to speculate in the Discussion (p.12) but need to go further, since it is not clear to this reviewer. Also, what is the significance of a 4-5-fold reduction in viral RNA levels at 4 days post infection?

Minor Issues.

1. While all Figure legends indicate the number of animals and statistical tests, none of the indicate how many independent experiments were performed. This should be added to each legend.
2. p.8, top and elsewhere. In the Results (not just the Methods) the authors should add the dose (in ug or mg/kg) and route of administration of proteins.
3. Figure 1h-i. This data on down-regulation of hACE2 is not altogether convincing...Less than 2-fold effect... Is it possible to stain tissues with anti-ACE2 that binds a different epitope? Indeed, in the top panel of panel h, the RBD-Fc looks like the sham.
4. Figure 1. Why is there no substantive deleterious effect of RBD-Fc and S1-Fc on lung injury? Why is it only seen in the context of acid-induced injury?
5. Histopathology. In Figures 2, 3, and 4, the authors show only single selected images of lung injury and effects of treatment. They should instead show low power images of the whole lung with high power insets. This will allow the reviewer to appreciate how representative the data is. Also, the gross pathology images in Figures 2 and 3 are helpful, as are the wet/dry ratio measurements. Can these analyses be performed in Figure 4?

6. p.9, Bottom and EDF 3d-e. The comment about improvement in renal function is not accurate, as no statistically significant benefit was observed. This should be edited.

7. Wouldn't soluble ACE2-Fc be a better drug than B38-CAP since it both inhibits SARS-CoV-2 entry and enhances conversion of angiotensin II? Perhaps some discussion about this point and the relative benefit to ACE inhibitors under clinical trial can be added. Why would B38-CAP be better than these other agents that disrupt the renin-angiotensin system?

8. p. 14. "Fonder" should be "Founder"

9. p. Histopathology scoring index. This description is not transparent. What exactly constitutes: 0 = minimal (little) damage, 1 = mild damage, 2 = moderate damage, 3= severe damage and 4 = maximal damage. More precise details of the pathological features at each stage is needed.

10. The RAS system maintains blood pressure homeostasis. To say it "worsens cardiovascular disease" is not accurate. Consider changing instead to something like, "ACE2 has been shown to be an important protective factor in several disease states including heart failure, myocardial infarction, hypertension, acute lung injury and diabetes."

11. Methods, viruses - "Live" SARS-CoV-2 should be changed to "infectious" SARS-CoV-2. Also, given the ongoing interest regarding SARS-CoV-2 tropism in relation to the furin cleavage site mutation, the authors should specify the passage history of the virus and confirm whether the furin cleavage site mutation is present in the stock used for this study, and if so, at what percent.

Reviewer #3 (Remarks to the Author):

In this well-written manuscript, the authors provide proof-of-principle from two independent preclinical models of SARS-CoV-2 induced lung injury demonstrating the protective effect of the ACE2-like enzyme B38-CAP in SARS-CoV-2 infection. The study is well performed and methods and controls are adequate. My main concerns relate to the potential mechanism of action of B38-CAP (reduced ATII levels vs. increased Ang-(1-7)), the proposed beneficial effects in extrapulmonary organs (which are of considerable clinical relevance, but presently underdeveloped), and the requirement for a two-hit model in the hamster.

Major comments:

1. The authors convincingly demonstrate the protective effect of B38-CAP in animal models of SARS-CoV-2 infection. However, it remains unclear whether this protection is caused by a reduction in ATII, an increased formation of Ang-(1-7), or both. This knowledge may have important clinical ramifications, as presently both ARBs and Ang-(1-7) are tested in clinical trials for the treatment of COVID-19. Ideally, the authors could test this in additional animal experiments using inhibitors of the Mas receptor in combination with B38-CAP, and inhibition of AT1 receptors in the absence of B38-CAP. At least, the authors should provide measures of plasma Ang-(1-7) levels.
2. In the hACE2 Tg mouse model cytokine levels and histological scores largely reflect the extent of the lung inflammatory response but not necessarily lung injury. As such, it would be important to show either some evidence of lung edema or impaired barrier function (e.g. W/D ratio, BALF protein, Evans Blue extravasation) or some indication of impaired physiological function (e.g. lung mechanics or oxygenation).
3. Notably, the effect of SARS-CoV-2 on ACE2 expression does not seem to be specific, but may reflect a general feature of acute lung injury (ALI) as it can be replicated by acid injury. This finding is in line with previous reports demonstrating reduced ACE2 expression and/or activity in various infectious or sterile models of ALI. This lack of specificity should be taken into consideration in the discussion.
4. In the hamster model, the effects of S1-Fc and RD-Fc were by themselves rather mild, and only caused severe lung injury when administered in a model of acid-induced lung injury. This scenario is different from the clinical course of COVID-19, which typically develops in otherwise healthy, uninjured lungs. This important difference should be discussed as a potential limitation of the study.
5. The authors provide data that markers in sera of RBD-Fc-treated hamsters indicate liver and kidney injury and show a convincing protective effect of B38-CAP treatment. Given the incidence of systemic (in particular renal) injury in clinical COVID-19, it would be interesting and relevant to

assess potential end-organ damage also at the macro- and microscopic level. As such, the addition of macro-/microscopic images would be extremely informative.

Minor comments:

1. In the result section, the authors state that “Treatment with S1-Fc or RBD-Fc downregulated expression of ACE2 in the lungs compared with control-Fc treatment (Fig. 1g-i).”, however, in Fig. 1i it does not seem that S1-Fc alone is significantly different from control.

2. Why are there no measurements of lung mechanics in the Acid+B38-CAP group in Fig. 3h,i?

3. In the extended data figure 2B (page 36) there is a band in the third lane cut off in the upper part of the ACE2 blot. Could the authors provide a larger image of this blot and explain what this band – which seems absent in the other lanes – may reflect? Is it possible that this is just an upward shift of the ACE2 band due to a posttranslational modification rather than a loss of ACE2?

4. The authors show that ACE2 activity was markedly increased in the plasma of B38-CAP-treated hamsters (page 9, line 176; figure 3C). As this measurement is likely not specific for ACE2 but also detects B38-CAP activity, this finding is probably little surprising following treatment with B38-CAP, and it remains unclear to which extent this result is attributable to restoration of endogenous ACE2 activity. This limitation should be clearly stated in the text.

5. The term “acid-induced swelling” of the lung is not adequate and should be changed to “acid-induced edema formation” or “acid-induced lung injury”.

6. Page 5, line 100: ACE2 Ref 26,27, please remove “Ref”

7. On page 18, line 390 the correct reference should be Imai et al., 2001 (PMID: 11568170) who first described this injury scoring system.

Response to the Reviews

Tomokazu Yamaguchi, et al. " **ACE2-like carboxypeptidase B38-CAP protects from SARS-CoV-2-induced lung injury** " (Manuscript number: NCOMMS-20-36848-T)

Point-by-point response to the reviewer comments

Reviewer #1

The authors present a manuscript where they aim to show and validate the use of B38-CAP, an enzyme with much similarity to ACE2, in instances of COVID-19.

Major Comments

• It has been shown (<https://doi.org/10.1038/s41586-020-2312-y>) that SARS-CoV-2 causes minimal injury/infiltration in WT mice compared to hACE2 transgenic mice, did you have similar observations between transgenic mice and non-transgenic wildtype controls? Were wildtype controls done for transgenic SARS-CoV-2 infection and rescue with B38-CAP?

Response: We thank the reviewer for this comment. We have conducted a new experiment to infect SARS-CoV-2 in wild type mice, to perform the requested control experiment. After virus inoculation, wild type mice showed neither body weight loss nor lethality and were not apparently infected with the virus, albeit viral RNA was detected in the lung in one out of three mice at 2 dpi (please see new Extended Data Fig. 6c-e). Virus-inoculated wild type mice exhibited neither lung edema nor pathological changes, whereas hACE2 transgenic mice showed severe lung diseases upon infection (Fig. 4d; Extended Data Fig. 7a, b). These results are consistent with previous work (<https://doi.org/10.1038/s41586-020-2312-y>), which described no apparent pathologies in SARS-CoV-2-inoculated wild type mice. On the other hand, Boudewijns R, Thibaut HJ, Neyts J, Dallmeier K, et al. have recently shown that wild type BALB/c mice but not C57BL/6 mice exhibit mild lung pathologies (<https://doi.org/10.1038/s41467-020-19684-y>). Because the genetic background of our hACE2 transgenic mice is C57BL/6, our data are in line with all the known literature we are aware of. We have now cited two references to better clarify SARS-CoV-2 infections of wild type mice; please see page 12; line 272 of the revised manuscript.

• Authors looked at western blot protein expression throughout paper, your non-specific banding is quite pronounced in some westerns and absent in others. Can a more specific antibody be used in place for some westerns?

Response: We tested other sources of anti-ACE2 antibodies for detection of hamster ACE2 protein in the lung lysates by Western blot, but the results are the same. In the hamster lung lysate samples, both “specific” ACE2 band and the “non-specific” band were decreased by SARS-CoV-2 infection or Spike plus acid-induced injury (Fig. 1b, c, j-m). In the pull down assays from hamster lung lysates, the upper “specific” ACE2 band was always bound to Spike proteins, while lower “non-specific” band did not associate with Spike (Fig. 1e). Thus, the

difference of the “specific” ACE2 band and “non-specific” band is the ability to bind to Spike. Of note, human ACE2 also showed an additional faint band below the main band, which was not co-precipitated with RBD-Fc or S1-Fc (Fig. 1e, left). Mammalian ACE2 is glycosylated for proper expression on the cell surface, and at least upper “specific” ACE2 band is the one properly folded and expressed on the cell surface thereby binding to Spike. Looking back our experience of analyzing ACE2 protein for nearly two decades, bands of endogenous ACE2 protein are always broad or sometimes appear as doublets depending on samples. For instance, in this study, the endogenous mouse ACE2 band appeared as doublets (please see Fig. 4e). In addition, immunohistochemistry of ACE2 protein detects ACE2-like immuno-reactivities inside the cells as well as cell at the membrane (DOI: 10.1152/ajpheart.00068.2005), implicating the presence of an unfolded or pre-glycosylated form of the protein. Thus, there is a possibility that lower “non-specific” band is unfolded or pre-glycosylated form of ACE2 protein. Recently, a short form of ACE2 has been identified that can be induced by interferon and lacks the N terminal domain and hence cannot bind to Spike (doi: 10.1038/s41588-020-00731-9.). Mass spec analysis might resolve the issue, which we hope can be done in the near future. We have modified the revised manuscript to state “unfolded or short form of ACE2 or non-specific band” in the figure legend.

• *Were controls conducted for ACE2 activity assay? If so please show in main figure or supplemental. Since plasma ACE2 activity was calculated based on B38-CAP activity and that there is no way to differentiate ACE2 and B38-CAP activity, how do we know that this bacterial enzyme is just not another variant of ACE2? Authors state in their seminal paper (<https://doi.org/10.1038/s41467-020-14867-z>) that B38-CAP is likely an evolutionary homologue of ACE2, and what has been presented in this paper seems to strongly corroborate that. Authors state that B38-CAP is preventing disease through cleaving Ang II into Ang 1-7, but don't show convincing enough evidence that B38-CAP isn't a simple homologue of ACE2, and similar to soluble rhACE2, binds to the coronavirus preventing disease onset to begin with. The included pull down assay is not convincing enough on its own especially in light of non-specificity in western blotting.*

Response: We thank the reviewer for this comment. We conducted control experiments with the ACE2 inhibitor MLN-4760, which also inhibits B38-CAP enzyme activity with the same efficiency as ACE2, and excluded non-specific activity in the assay by subtracting the values measured with MLN-4760 from the values without MLN-4760 (Extended Data Fig. 4a). Thus, ACE2 activity detected in the plasma of B38-CAP-treated animals indeed represents injected B38-CAP. We have amended the figure legend for better understanding.

As for potential binding of B38-CAP to the coronavirus preventing disease onset, we conducted a new experiment of SARS-CoV-2 infection in the cell culture to examine the effects of B38-CAP on cell entry of the virus *in vitro*. The results confirm that B38-CAP does not inhibit the infection of SARS-CoV-2 in Vero E6/TMPRSS2 cells in the culture (please see new

Extended Data Fig. 3c, d), supporting the notion that B38-CAP does not bind to Spike of SARS-CoV-2.

• *Why did you test transgenic mice/acid-induced lung injury hamsters with only the RBD-Fc?*

Response: We thank the reviewer for this comment. We have now also tested trimeric Spike protein (Spike-6P) plus acid-induced lung injury hamsters for B38-CAP treatment. Our new data confirm the therapeutic effects of B38-CAP on Spike plus acid-induced lung injury (please see new data in Figure 2).

Minor comments:

• *Your mice/hamsters also had minimal injury with virus in the absence of acid inhalation, is this due to increased ACE2 protein expression with acid treatment? And could increase ACE2 expression due to the acid be leading to more viral infiltration and exuberating of disease severity?*

Response: Acid aspiration decreases ACE2 expression in the lungs of hamsters (Figure 1j, k) and wild type mice (doi: 10.1038/nature03712). As for the case in viral infection without acid inhalation, we demonstrated that SARS-CoV-2 infection downregulates ACE2 expression in the lungs of hamsters (Figure 1b, c) in our original submission. In this revision, we have further examined expression of human ACE2 and mouse Ace2 in the lungs of hACE2 transgenic mice infected with SARS-CoV-2, and found that bulk protein expression of both human ACE2 and mouse Ace2 were significantly downregulated by SARS-CoV-2 infection (Figure 4e-g). Since supplementation of ACE2 activity with B38-CAP treatment improved the disease severity in hamsters and hACE2 transgenic mice (Figure 3e-h, 4l-p), our data indicate that decreased ACE2 expression is at least in part involved in the pathogenesis of lung injury upon SARS-CoV-2 infection.

Regarding the question about effects of acid plus virus inoculation in the lungs: since acid aspiration decreases ACE2 expression in the lungs, it is conceivable that subsequent inoculation of SARS-CoV-2 might decrease viral infiltration in the lungs.

• *Be specific on whether you are talking about protein or gene expression throughout the manuscript, please.*

Response: Our apologies for this oversight. We now clearly specify protein or mRNA expression throughout the manuscript.

• *Include control image for lung pathology (Figure 4D).*

Response: We now included a control image for lung pathology (new Figure 4m).

• *No significance shown for decrease in ACE2 (Figure 1I) between control-Fc and S1-Fc (lines 142-143). Please include control-Fc sham group label in Figures.*

Response: We have added P value for comparison between control-Fc and S1-Fc (new Figure 1k).

• *In the abstract (lines 45-47) you say that spike protein and the RBD directly elevated Ang II levels when this was not the case (Figure 1J).*

Response: We thank the reviewer for this comment. We modified the text in the abstract (page 3; lines 56-58).

• *Authors claim to be investigating discrepancy between loss of ACE2 with viral infection versus reports of increased ACE2 mRNA in specific lung cell types, authors need to show mRNA levels and whether they track with their protein expression data to make a conclusive point (lines 86-87).*

Response: We thank the reviewer for this comment. We have measured bulk ACE2 mRNA in the lungs of SARS-CoV-2-infected hamsters, showing that mRNA levels of ACE2 were downregulated by viral infection (Extended Data Fig. 1a), consistent with decreased ACE2 protein abundance (Fig. 1b, c).

• *It is of note that spike proteins downregulated ACE2, but did not change RAS activation (Ang II), in the absence of acid injury, which ties in with that COVID-19 severity is primarily for those with underlying comorbidities (Figure 1J).*

Response: We have now stated that Spike proteins downregulated ACE2 and activated the RAS in the presence of acid aspiration-induced lung injury but not in the absence of lung injury (page 8; lines 152-159).

Reviewer #2

Review of Minato et al

This paper evaluates the contribution of Angiotensin-converting enzyme 2 (ACE2) activity to the pathogenesis of SARS-CoV-2-induced lung injury using hamster and mouse models. In addition to being the entry receptor for SARS-CoV-2, ACE2 is the carboxypeptidase that degrades angiotensin II to angiotensin 1-7; this attenuates the renin-angiotensin system, which has been shown previously to reduce blood pressure, cardiovascular disease and acute lung injury. Here, the authors perform a series of experiments to address whether the carboxypeptidase enzymatic activity of ACE2 protects against lung injury in hamsters that is induced by SARS-CoV-2 proteins (RBD and S), which downregulate expression of ACE2, or the virus itself in the context of challenge in human ACE2 transgenic mice (expressed under the CAG promoter). Administration of S1-Fc or RBD-Fc directly downregulated ACE2 expression in vivo and resulted in elevated angiotensin II levels and worsened acid-induced lung injury in hamsters. Treatment with B38-CAP, a recombinant protein with ACE2 activity that does not bind S or RBD, downregulated the S1-Fc or RBD-Fc induced increase in angiotensin II, and prevented pulmonary inflammation and disease. The authors went on to test the efficacy of B38-CAP in SARS-CoV-2-infected hACE2 transgenic mice. When B38-CAP was administered starting 12 hours after SARS-CoV-2 infection, lung inflammation and pathology showed improvement at 4 days post-infection (dpi), and this was associated with a small reduction in viral RNA levels even though B38-CAP does not bind S or RBD. The authors conclude that this paper demonstrates that increasing ACE2 enzymatic activity can be a therapeutic strategy for COVID-19 independent of engagement of the spike protein (which is the basis for some soluble ACE2-Fc therapeutic approaches).

The strengths of this generally easy to follow paper include the quality of the data, the novelty of the approach and inhibitor, the use of two independent animal models, and its potential for translation either directly or indirectly (i.e., using soluble ACE2 with active enzymatic activity or existing drugs that block angiotensin II receptors or inhibit ACE activity). The weaknesses of the study relate to the use of S1-Fc and RBD-Fc and the differences compared to the proteins on virions, the timing of clinical analysis in the hACE2 transgenic mice, and the noticeable absence of virus protection studies in hamsters despite information in the Methods. That said, a few additional controls along with experiments especially in the COVID-19 model could provide needed support for the conclusions of the paper and the potential utility of targeting the renin-angiotensin axis during SARS-CoV-2 infection.

Major Issues.

1. Effects of S1-Fc and RBD-Fc. The authors show in Figure 2 and 3 that administration of S1-Fc and RBD-Fc to hamsters results in down-regulation of ACE2 receptor and increased angiotensin II levels that then correlate with acute lung injury. One major difference between

these recombinant proteins and the S on native SARS-CoV-2 virions is the Fc moiety, which could engage immune cells to independently mediate damage and or injury. To confirm these phenotypes are not artifacts of immune cell engagement, the authors should consider generating a LALA-PG or N297Q variant that cannot engage C1q or Fc- gamma receptors and show it mediates the same disease enhancing effects of the parental WT S1-Fc or RBD-Fc. This would also seem important because most advanced vaccine platforms or S protein-based. Is there any evidence from early clinical trials of the vaccines exacerbating lung injury in those with pre-existing pulmonary conditions?

Response: We thank the reviewer for this comment. As suggested by the reviewer, we tried but failed to generate LALA-PG or N297Q variant of S1-Fc and RBD-Fc. Instead, we have prepared Fc-free trimeric Spike protein with 6 proline substitutions and prefusion conformation (Spike-6P) (new Fig. 1d; Extended Data Fig. 1c, f, g) (doi: 10.1126/science.abd0826) and tested it in hamsters. Spike-6P treatment downregulated ACE2 expression in the lungs of hamsters with acid aspiration (new Figure 1l, m), increased Angiotensin II levels (Fig. 2c) and exacerbated lung injury (Figure 2d-g). Thus, we have excluded the possibility that Fc moiety is solely responsible for the exacerbation of lung injury. For proper discussion, we have mentioned potential Fc moiety effects in the Results section (pages 8; lines 173-175).

In relation to the vaccines, while there is no evidence that mRNA vaccines for SARS-CoV-2 exacerbate lung injury in those with pre-existing pulmonary conditions, SARS-CoV vaccine-associated eosinophil accumulation in the respiratory tract has been reported in preclinical immunization studies. Nevertheless, vaccination is not done for patients with pneumonia or other inflammatory lung diseases, and the amount of mRNA vaccine-generated Spike protein in the body appears to be lower than in our experiments. We have included this in the Discussion section (pages 16; lines 342-344). However, it might be indeed interesting to test whether the currently approved vaccines might actually reduce ACE2 expression following immunization of *e.g.* mice or hamsters, also in light of emerging side effects such as increased incidences of myocarditis in young males.

2. Hamster challenge with SARS-CoV-2. The authors perform all preliminary studies with S1-Fc and RBD-Fc in hamsters and yet do not show treatment and challenge data with B38-CAP in these animals. Based on the description in the Methods section (p.15, middle) it appears that hamsters were indeed infected with 10e3 PFU of SARS-CoV-2 UT-NCGM02) and 10e5 PFU of SARS-CoV-2 HKU-001a. Where is this data? This should be included as it directly links to the studies in Figures 2 and 3.

Response: Our original submission did not have data on B38-CAP treatment in SARS-CoV-2 infected hamsters due to limited access to the BSL3 laboratory at that time. We have now conducted the requested experiments of B38-CAP treatment in SARS-CoV-2 infected hamsters. Importantly, our results show that B38-CAP treatment mitigated lung injury induced by SARS-CoV-2 infections in hamsters (please see new Fig. 3; Extended Data Fig. 5).

3. CAG-hACE2 transgenic mice. In Figure 4, the authors treated SARS-CoV-2 infected transgenic mice with B38-CAP beginning at 12 hours post-infection and then harvested tissues at day+4. There are several issues: (a) The CAG-hACE2 transgenic mice have not been described for SARS-CoV-2 infection and pathogenesis. While the authors allude to a paper in preparation (p.14, bottom), for the reader to understand this model, some data needs to be added to this paper (e.g., weight loss and survival; kinetics of viral infection in lungs). (b) The choice of pathological and immunological evaluation for a drug at day+4 is unusual given that many other hACE2-transgenic mice show worse disease at later time points (e.g., day+7 or +8 [PMID:32841215, 32839612, 32516571]). What happens later? Does B38-CAP prevent lethality or severe clinical disease in this CAG-hACE2 transgenic mice? This is an important test of a drug. (c) A more thorough investigation of disease measures, as exemplified by Figure 3, is warranted in the COVID animal model. Some additional measures would greatly enhance the conclusions.

Response: We thank the reviewer for this comment. As suggested by the reviewer, we have analyzed SARS-CoV-2 infection and pathogenesis in CAG-hACE2 Tg mice, compared to wild type control mice. All the infected CAG-hACE2 Tg mice showed decreased body weights and had died by 9 days after infection, whereas - as expected - the virus-inoculated wild type mice appeared healthy during the entire observation period (please see Extended Data Fig. 6a-d). High viral titer and high copy numbers of viral RNA in the lungs peaked at 2 dpi and decreased until day 7 in the infected CAG-hACE2 Tg mice (Fig. 4b; Extended Data Fig. 6e). Lung edema and pathologies of lung injury peaked on day 2-4, but lung injury and pulmonary cytokine levels were markedly downregulated through day 7 (Fig. 4d; Extended Data Fig. 7a-c). Viral replication was the highest in brain among the organs examined and peaked at day 7; in addition brain cytokine levels were markedly increased at day 7 (Fig. 4c; Extended Data Fig. 6f; Extended Data Fig. 7d). Altogether, SARS-CoV-2 infection in the CAG-hACE2 Tg mice induces severe lung injury until 4 dpi, while the mice likely die of explosive viral replication and inflammation in the brain. It is thus not possible to evaluate SARS-CoV-2 induced lung injury at days 7 or 8 in this CAG-hACE2 Tg mice, and hence we have examined the effects of B38-CAP on lung injury at 4 dpi (Fig. 4h). These infection studies and B38-CAP treatment were repeated three times with consistent results. Importantly, to more thoroughly assess lung pathologies, for the revised paper we have set up and analyzed respiratory function and lung edema as well as various aspects of lung pathologies within the BSL3 laboratory. All of these data have now been included in the revised manuscript (Fig. 4l-p).

4. In Figure 4c, the authors show that B38-CAP appears to lower viral burden. How does this work since it neither binds to S or RBD? Does B38-CAP down-regulate ACE2 expression through some type of negative feedback? The authors start to speculate in the Discussion

(p.12) but need to go further, since it is not clear to this reviewer. Also, what is the significance of a 4-5-fold reduction in viral RNA levels at 4 days post infection?

Response: We thank the reviewer for this comment. As suggested by the reviewer, we have repeated SARS-CoV-2 infections and pathogenesis studies in CAG-hACE2 Tg mice, as mentioned above. Our data indicate that B38-CAP does not affect viral replication in hamsters and CAG-hACE2 Tg mice (Fig. 3c, 4j).

Minor Issues.

1. While all Figure legends indicate the number of animals and statistical tests, none of the indicate how many independent experiments were performed. This should be added to each legend.

Response: Our apologies for this omission. The numbers of independent experiments have now been added to each figure legends.

2. p.8, top and elsewhere. In the Results (not just the Methods) the authors should add the dose (in ug or mg/kg) and route of administration of proteins.

Response: The dose and route of administration of proteins has been added in the revised text of the Results section.

3. Figure 1h-i. This data on down-regulation of hACE2 is not altogether convincing...Less than 2-fold effect... Is it possible to stain tissues with anti-ACE2 that binds a different epitope? Indeed, in the top panel of panel h, the RBD-Fc looks like the sham.

Response: We thank the reviewer for this comment. The data of Figure 1h-i in the original submission (Fig. 1j, k in the revision) show down-regulation of hamster ACE2 but not hACE2 (human ACE2). We realize that ACE2 downregulation is modest in response to RBD-Fc or S1-Fc in the absence of lung injury. We tested other sources of ACE2 antibodies, and the results were the same. When we injected Spike-6P to hamsters, ACE2 expression was downregulated in the hamsters with acid lung injury but not in the absence of lung injury (Fig. 1l, m; Extended Data Fig. 1h, i). In our cell culture experiments, Spike treatment *per se* clearly downregulated cell-surface expression of ACE2 (Fig. 1f, g), similar to SARS-CoV Spike-mediated ACE2 downregulation (Kuba K, *Nat Med* 2005; Wang H, *Cell Res* 2008). It should be noted that Western blot analysis quantifies intracellular ACE2 protein as well as ACE2 protein on the cell surface, but only cell surface bound ACE2 (and shed ACE2) catalyze the degradation of Angiotensin II peptide. Importantly, Spike treatment plus acid inhalation downregulates the abundance of ACE2 protein in the lungs, thereby upregulating Angiotensin II levels and exacerbating lung injury *in vivo*, whereas *in vitro* Spike itself can downregulate cell-surface ACE2 expression. Please see new text on page 8; lines 162-164, to clarify this issue.

4. Figure 1. Why is there no substantive deleterious effect of RBD-Fc and S1-Fc on lung injury? Why is it only seen in the context of acid-induced injury?

Response: Spike protein downregulates ACE2 expression on the cell surface *in vitro*, and Spike treatments downregulate ACE2 abundance in the lungs with acid-induced injury *in vivo*. It is well known that only ACE2 deficiency or treatment with ACE2 inhibitors does not induce lung injury, whereas ACE2 down-regulation exacerbates lung injury induced by another insult, such as acid aspiration, LPS challenge, or viral infection, as multiple publication has shown worse outcomes of acute lung injury due to ACE2 downregulation (Imai Y, *Nature* 2005; Rey-Parra GJ, *J Mol Med* 2012; Zou Z, *Nat Commun* 2014). This is of course an important issue that needs careful consideration, because if Spike protein induces lung injury by itself, the current vaccines might also have the risk to trigger side effects via ACE2.

5. Histopathology. In Figures 2, 3, and 4, the authors show only single selected images of lung injury and effects of treatment. They should instead show low power images of the whole lung with high power insets. This will allow the reviewer to appreciate how representative the data is. Also, the gross pathology images in Figures 2 and 3 are helpful, as are the wet/dry ratio measurements. Can these analyses be performed in Figure 4?

Response: As requested by the reviewer, we now show low power images of the lung histology in Figures 2-4. In addition, we conducted wet/dry ratio measurements in experimental infection of hamsters and CAG-hACE2 Tg mice (please see new Fig. 3f, 4l).

6. p.9, Bottom and EDF 3d-e. The comment about improvement in renal function is not accurate, as no statistically significant benefit was observed. This should be edited.

Response: We removed the comment about improvement in renal function and simply mentioned no apparent toxicities of B38-CAP detected in kidney and liver as defined by the markers tested (page 10; lines 221-222).

7. Wouldn't soluble ACE2-Fc be a better drug than B38-CAP since it both inhibits SARS-CoV-2 entry and enhances conversion of angiotensin II? Perhaps some discussion about this point and the relative benefit to ACE inhibitors under clinical trial can be added. Why would B38-CAP be better than these other agents that disrupt the renin-angiotensin system?

Response: We completely agree that the dual effects of soluble ACE2 might be better than a single effect of B38-CAP. We performed our study because it is critical to experimentally dissect these two functions. As for comparison to ACE inhibitors, blockade of the renin-angiotensin system with already clinically available small compounds would be better than a recombinant protein like soluble ACE2 or B38-CAP in terms of drug development. However, the enzymatic activity of ACE2 or B38-CAP also mediates the conversion of des-Arg⁹-bradykinin and apelin in addition to angiotensin II, which may provide further beneficial effects to improve lung injuries and systemic tissue damage in COVID-19. We have now included

these important considerations to the Discussion section of the revised manuscript (pages 16; lines 357-369).

8. p. 14. “Fonder” should be “Founder”

Response: We have corrected this mistake.

9. p. Histopathology scoring index. This description is not transparent. What exactly constitutes: 0 = minimal (little) damage, 1 = mild damage, 2 = moderate damage, 3= severe damage and 4 = maximal damage. More precise details of the pathological features at each stage is needed.

Response: Apologies for this oversight. We now present what parameters are changed in Extended Data Tables 2-5. We have also stated this clearly in the main text.

10. The RAS system maintains blood pressure homeostasis. To say it “worsens cardiovascular disease” is not accurate. Consider changing instead to something like, “ACE2 has been shown to be an important protective factor in several disease states including heart failure, myocardial infarction, hypertension, acute lung injury and diabetes.”

Response: We thank the reviewer for this suggestion. We accordingly modified the text (page 4; lines 84-85).

11. Methods, viruses - “Live” SARS-CoV-2 should be changed to “infectious” SARS-CoV-2. Also, given the ongoing interest regarding SARS-CoV-2 tropism in relation to the furin cleavage site mutation, the authors should specify the passage history of the virus and confirm whether the furin cleavage site mutation is present in the stock used for this study, and if so, at what percent.

Response: We thank the reviewer for this comment. We corrected “live” to “infectious” SARS-CoV-2 in the Methods. We sequenced the furin cleavage site in RNA samples from the infected animal lungs and detected no mutations in the furin cleavage site. We specified the passage history of the virus and added to Methods section that we failed to detect mutations in the furin cleavage site (please see page 18; lines 383-402).

Reviewer #3

In this well-written manuscript, the authors provide proof-of-principle from two independent preclinical models of SARS-CoV-2 induced lung injury demonstrating the protective effect of the ACE2-like enzyme B38-CAP in SARS-CoV-2 infection. The study is well performed and methods and controls are adequate. My main concerns relate to the potential mechanism of action of B38-CAP (reduced ATII levels vs. increased Ang-(1-7)), the proposed beneficial effects in extrapulmonary organs (which are of considerable clinical relevance, but presently underdeveloped), and the requirement for a two-hit model in the hamster.

Major comments:

1. The authors convincingly demonstrate the protective effect of B38-CAP in animal models of SARS-CoV-2 infection. However, it remains unclear whether this protection is caused by a reduction in ATII, an increased formation of Ang-(1-7), or both. This knowledge may have important clinical ramifications, as presently both ARBs and Ang-(1-7) are tested in clinical trials for the treatment of COVID-19. Ideally, the authors could test this in additional animal experiments using inhibitors of the Mas receptor in combination with B38-CAP, and inhibition of AT1 receptors in the absence of B38-CAP. At least, the authors should provide measures of plasma Ang-(1-7) levels.

Response: We thank the reviewer for this important comment. We have now included Ang-(1-7) measurements using ELISA. We first measured plasma Ang-(1-7) levels in hamsters treated with B38-CAP and found that B38-CAP treatment modestly upregulated Ang-(1-7) levels in the hamsters with only acid- or Spike plus acid-induced lung injury (please see new Extended Data Fig. 4c). We next measured Ang-(1-7) and ATII (Ang II) levels in the lung lysates of hACE2 Tg mice infected with SARS-CoV-2 and treated with B38-CAP, but Ang-(1-7) peptides were not detectable in our ELISA system (not shown, page 13; lines 284-285). On the other hand, Ang II levels were significantly downregulated by B38-CAP treatment (new Fig. 4k). Since B38-CAP does not further convert Ang-(1-7), B38-CAP-generated Ang-(1-7) is presumed to be metabolized to small fragment peptides by other enzymes in our lung injury models. This data indicates that B38-CAP protects from lung injury by a reduction in Ang II levels rather than an increased formation of Ang-(1-7). It would be certainly interesting to test inhibitors of the Mas receptor in combination with B38-CAP, but this requires large number of animals and further access to BSL3 facility. We would like to do these experiments as a new project if acceptable.

2. In the hACE2 Tg mouse model cytokine levels and histological scores largely reflect the extent of the lung inflammatory response but not necessarily lung injury. As such, it would be important to show either some evidence of lung edema or impaired barrier function (e.g. W/D ratio, BALF protein, Evans Blue extravasation) or some indication of impaired physiological function (e.g. lung mechanics or oxygenation).

Response: We have now set up and measured lung edema and lung functions within the BSL3 laboratory. In SARS-CoV-2 infected hamsters and hACE2 Tg mice, the W/D ratio was elevated with a peak at 4 days post infection (please see new Fig. 3f, 4d; Extended Data Fig. 5c). In addition, we measured lung function in SARS-CoV-2-infected hACE2 Tg mice at 4 days after infection, and observed significant upregulation of elastance and resistance (new Fig. 4o, p) confirming that lung injury was indeed induced by SARS-CoV-2 infection. Importantly, using these readout, B38-CAP treatment suppressed SARS-CoV-2-induced lung injury (Fig. 3e-h, 4l-p; Extended Data Fig. 5-7).

3. Notably, the effect of SARS-CoV-2 on ACE2 expression does not seem to be specific, but may reflect a general feature of acute lung injury (ALI) as it can be replicated by acid injury. This finding is in line with previous reports demonstrating reduced ACE2 expression and/or activity in various infectious or sterile models of ALI. This lack of specificity should be taken into consideration in the discussion.

Response: We thank the reviewer for pointing out this important aspect of ACE2 regulation. Spike protein downregulates ACE2 expression from the cell surface *in vitro*, and treatment with Spike downregulates ACE2 abundance in the lungs with acid-induced injury *in vivo*. We thus speculate that downregulation of ACE2 during SARS-CoV-2 infection is regulated by two mechanisms: 1. Spike binding or viral entry-mediated specific downregulation, and 2. non-specific downregulation due to ALI and inflammation induced by viral infection, as the reviewer correctly pointed out. For mechanistic insights into the latter, inflammatory signals such as NF- κ B activation have been shown to downregulate ACE2 expression (doi:10.1038/celldisc.2017.21). We have included these considerations into the discussion of the revised manuscript (pages 15; lines 327-341).

4. In the hamster model, the effects of S1-Fc and RD-Fc were by themselves rather mild, and only caused severe lung injury when administered in a model of acid-induced lung injury. This scenario is different from the clinical course of COVID-19, which typically develops in otherwise healthy, uninjured lungs. This important difference should be discussed as a potential limitation of the study.

Response: We agree that there is a potential limitation of the experiments using Spike protein (S1-Fc or RBD-Fc) plus acid induced lung injury. On the other hand, the pathogenesis of COVID-19 contains many complicated mechanisms, and we believe that the virus-free experimental model of Spike plus acid induced lung injury will be useful to dissect the mechanisms which are associated with conjugation of Spike and ACE2 required for viral entry. We have now added the potential limitation of the experiments to the revised text (pages 15; lines 333-336).

5. The authors provide data that markers in sera of RBD-Fc-treated hamsters indicate liver and kidney injury and show a convincing protective effect of B38-CAP treatment. Given the incidence of systemic (in particular renal) injury in clinical COVID-19, it would be interesting and relevant to assess potential end-organ damage also at the macro- and microscopic level. As such, the addition of macro-/microscopic images would be extremely informative.

Response: We thank the reviewer for this valuable suggestion. However, the serum markers of kidney, unlike the liver injury markers, were not significantly elevated because of large individual variations (Extended Data Fig. 4I). We have now carefully examined the kidney histopathology of those hamsters and consistently observed variations in severity of renal injury; some animals exhibited glomerular damage and congestion/hemorrhage, while others showed little or no pathological changes. Thus, unfortunately, it appears that our Spike plus acid-induced lung injury model is not well suited to evaluate end-organ damage at this moment. In a future project, we would like to develop another animal model of COVID-19-associated end-organ damage to experimentally address this critical question.

Minor comments:

1. In the result section, the authors state that “Treatment with S1-Fc or RBD-Fc downregulated expression of ACE2 in the lungs compared with control-Fc treatment (Fig. 1g-i).”, however, in Fig. 1i it does not seem that S1-Fc alone is significantly different from control.

Response: We thank the reviewer for pointing this out. We corrected this sentence to the statement that treatment with Spike protein (S1-Fc or RBD-Fc) downregulates ACE2 abundance in lungs with acid-induced injury but not in the absence of lung injury in the results section (page 8; lines 152-162).

2. Why are there no measurements of lung mechanics in the Acid+B38-CAP group in Fig. 3h,i?

Response: For measurements of lung mechanics, hamsters were not available again for setting up a new acid+B38-CAP experimental cohort, because high demand for SARS-CoV-2 studies limited our access to the infectious units and ventilators. Our sincere apologies but, in essence, because of other priorities of the company that performed these experiments, we were not allowed access anymore.

3. In the extended data figure 2B (page 36) there is a band in the third lane cut off in the upper part of the ACE2 blot. Could the authors provide a larger image of this blot and explain what this band – which seems absent in the other lanes – may reflect? Is it possible that this is just an upward shift of the ACE2 band due to a posttranslational modification rather than a loss of ACE2?

Response: We thank the reviewer for this comment. The Western blot of Extended Data Figure 2B in original submission (Extended Data Figure 3b in this revision) shows *in vitro* binding of ACE2 to RBD-Fc. The band in the third lane is a protein non-specifically precipitated with control-Fc. Our apologies if that was unclear from the figure legend. Nevertheless, it would be indeed very interesting to test if ACE2 downregulation in lung injury is regulated by post-translational modifications of ACE2.

4. The authors show that ACE2 activity was markedly increased in the plasma of B38-CAP-treated hamsters (page 9, line 176; figure 3C). As this measurement is likely not specific for ACE2 but also detects B38-CAP activity, this finding is probably little surprising following treatment with B38-CAP, and it remains unclear to which extent this result is attributable to restoration of endogenous ACE2 activity. This limitation should be clearly stated in the text.

Response: As suggested by the reviewer, we have now moved the data of ACE2 activity measurements in the plasma of B38-CAP-treated hamsters (Figure 3c in the original submission) to Extended Data Fig. 4a. Certainly, measured ACE2 activity may contain the activity of endogenous soluble ACE2 in addition to injected B38-CAP, but soluble ACE2 is not expressed by alternative splicing in transcription but solely generated by post-translational mechanisms, and as far as we know, B38-CAP does not induce shedding of ACE2 from the cell surface. Thus, measured ACE2 activity primarily represents injected B38-CAP in circulation. To make sure and state this limitation, we have reworded the Results section (pages 10; lines 199-204).

5. The term “acid-induced swelling” of the lung is not adequate and should be changed to “acid-induced edema formation” or “acid-induced lung injury”.

Response: Apologies for the loose language. We corrected the term to “acid-induced edema formation”.

6. Page 5, line 100: ACE2 Ref 26,27, please remove “Ref”

Response: We removed “Ref”.

7. On page 18, line 390 the correct reference should be Imai et al., 2001 (PMID: 11568170) who first described this injury scoring system.

Response: We thank the reviewer for this comment. We replaced it with the correct reference (PMID: 11568170).

REVIEWERS' COMMENTS

Reviewer #1 (Remarks to the Author):

My comments have been addressed

Reviewer #2 (Remarks to the Author):

The authors have added a substantial amount of new data that addresses the major concerns of this Reviewer. The paper is now strengthened and will be an important contribution that stimulates further research and possibly clinical trials.

The authors should consider editing by a native English speaker as parts of the paper require modification for enhanced clarity and diction.

Reviewer #3 (Remarks to the Author):

The authors have adequately responded to all my comments. I have no further concerns.

Response to the Reviews

Tomokazu Yamaguchi, et al. " **ACE2-like carboxypeptidase B38-CAP protects from SARS-CoV-2-induced lung injury** " (Manuscript number: NCOMMS-20-36848A)

Point-by-point response to the reviewer comments

Reviewer #1

My comments have been addressed

Response: We thank the reviewer very much for carefully and constructively reviewing our manuscript.

Reviewer #2

The authors have added a substantial amount of new data that addresses the major concerns of this Reviewer. The paper is now strengthened and will be an important contribution that stimulates further research and possibly clinical trials.

The authors should consider editing by a native English speaker as parts of the paper require modification for enhanced clarity and diction.

Response: We thank the reviewer very much for carefully and constructively reviewing our manuscript. According to your suggestion, we asked a native English speaker edit the manuscript.

Reviewer #3

The authors have adequately responded to all my comments. I have no further concerns.

Response: We thank the reviewer very much for carefully and constructively reviewing our manuscript.